# Gene signature discovery and systematic validation across diverse clinical cohorts for TB prognosis and response to treatment

**Roger Vargas**[1,2], **Liam Abbott**[1], **Daniel Bower**[1], **Nicole Frahm**[1], **Mike Shaffer**[1], **Wen-Han Yu**[1]*

**1** Bill & Melinda Gates Medical Research Institute, Cambridge, Massachusetts, United States of America,
**2** Harvard University, Cambridge, Massachusetts, United States of America

* wenhan.yu@modernatx.com

**Data Availability Statement:** All data used in this report were collected from publicly available databases (NCBI GEO and EMBL ArrayExpress) and the list of the accession ID of the datasets are

## Abstract

While blood gene signatures have shown promise in tuberculosis (TB) diagnosis and treatment monitoring, most signatures derived from a single cohort may be insufficient to capture TB heterogeneity in populations and individuals. Here we report a new generalized approach combining a network-based meta-analysis with machine-learning modeling to leverage the power of heterogeneity among studies. The transcriptome datasets from 57 studies (37 TB and 20 viral infections) across demographics and TB disease states were used for gene signature discovery and model training and validation. The network-based meta-analysis identified a common 45-gene signature specific to active TB disease across studies. Two optimized random forest regression models, using the full or partial 45-gene signature, were then established to model the continuum from *Mycobacterium tuberculosis* infection to disease and treatment response. In model validation, using pooled multi-cohort datasets to mimic the real-world setting, the model provides robust predictive performance for incipient to active TB risk over a 2.5-year period with an AUROC of 0.85, 74.2% sensitivity, and 78.3% specificity, which approximates the minimum criteria (>75% sensitivity and >75% specificity) within the WHO target product profile for prediction of progression to TB. Moreover, the model strongly discriminates active TB from viral infection (AUROC 0.93, 95% CI 0.91–0.94). For treatment monitoring, the TB scores generated by the model statistically correlate with treatment responses over time and were predictive, even before treatment initiation, of standard treatment clinical outcomes. We demonstrate an end-to-end gene signature model development scheme that considers heterogeneity for TB risk estimation and treatment monitoring.

## Author summary

Developing new diagnostic tools is one of the key areas to accelerate progress towards TB eradication. Having an accurate and rapid molecular test for TB detection in early stage has been highlighted in the recent updates of WHO guidelines and that facilitates faster treatment and reduces the risk of disease transmission. Blood gene signatures have shown

available on S1 Table. All the code for statistical modeling and figure visualization can be accessed on GitHub (https://github.com/wenhan-yu/tb-common-gene-signature).

**Funding:** The author(s) received no specific funding for this work.

**Competing interests:** The authors have declared that no competing interests exist.

promise in TB diagnosis, however, early detection and robust diagnosis against diverse populations are still the challenges to be overcome. Here we present a new computational approach leveraging the power of diverse clinical cohorts that not only defines a common gene signature specific to active TB disease but establishes a generalized predictive model. Importantly, we demonstrate robust performance of the model in both short-term and long-term TB risk estimation that provides a complementary approach to the current models, most of which offer good performance in active TB diagnosis and/or short-term TB risk estimation. In addition, we also demonstrate the utility of the model to monitor treatment responses along with *Mycobacterium tuberculosis* elimination, which provides additional information for evaluation before the end of treatment.

## Introduction

Tuberculosis (TB) is a leading cause of death globally from the infectious disease agent, *Mycobacterium tuberculosis* (*Mtb*), causing an estimated 10 million new cases of disease and 1.3 million deaths per year [1]. About 85% of individuals who develop active TB disease (ATB) can be successfully treated, but not all cases are diagnosed [1]. Better diagnostic tools are necessary to reach the WHO 2035 targets of a 95% reduction in global TB deaths and a 90% reduction in global TB incidence (compared to the 2015 baseline), in addition to developing new interventions for TB prevention and treatment [1]. An accurate, rapid, point-of-care (PoC), non-sputum-based diagnostic test is required in low and middle-income countries to identify individuals with ATB who require treatment, and to predict which individuals with latent tuberculosis infection (LTBI) are likely to progress to ATB, and therefore require preventive therapy [2–4]. Further, such a test may be used to monitor treatment response in individuals who commence TB drug treatment to ensure that they are cured at the end of treatment, or to allow for treatment modification if cured before the completion of the standard 6-month treatment regimen [1].

Several whole blood diagnostic gene signatures based on host immune responses have demonstrated the ability to accurately distinguish between ATB and healthy controls (HC), as well as ATB and disease manifestations such as other lung diseases (OLD) or LTBI [3,5–16]. Further, some studies have also demonstrated the use of gene signatures in predicting progression from LTBI to ATB [3,5,11,13,16–29] and in monitoring treatment response [3,8,16,30–35]. For TB gene signature discovery, most studies utilized a conventional approach beginning with a single cohort with differential gene expression analysis between ATB and different clinical conditions (LTBI, OLD or HC), followed by multivariate modeling to define the gene signature. The limited sample size in a single cohort, which is substantially smaller than the number of measured genes, increases the risk of multicollinearity and model overfitting. As a result, the conventional approach often insufficiently captures cohort heterogeneity and therefore generates a gene signature specific to the cohort examined. A systematic comparison of 16 published gene signatures, most of which compared cohorts derived from one location, demonstrated a limited overlap of the genes among these signatures [2]. Whether this stems from limited heterogeneity in single cohort samples or due to the diversity in the analytical approaches remains unclear.

In addition to diverse immune responses associated with different states in a wide spectrum of *Mtb* infectious pathophysiology [36], heterogeneity in cohort studies due to demographics, disease comorbidities, sample collection (whole blood vs. PBMCs) and transcriptomic profiling technologies (qRT-PCR, microarray, or RNAseq) make it challenging to translate the use

of any one gene signature into a generalizable PoC diagnostic test for use in real-world patient populations [2,18,37]. Among all existing gene signatures, two models (Sweeney 3 and RISK 6) demonstrated their utility as a screening or triage test for TB diagnosis in multicenter prospective studies [32,38], which showed potential to be more generalizable for different patient populations. To integrate population heterogeneity as part of the model building process, we conducted a meta-analysis where the transcriptome datasets from 37 published studies, all of which contained ATB and at least one other disease conditions (LTBI, OLD, or HC), were pooled together to capture various confounding factors inherited from individual studies in order to mimic a real-world setting. The distributions of TB scores in Sweeney 3 and RISK 6, when applied to this pooled dataset, were shown to be less distinguishable between ATB and other disease conditions than when applied in their original populations. This poses a challenge for calling a positive case from a screening or triage test for disease diagnosis due to the impact of patient and population level heterogeneity.

To this end, we aimed to assess whether we could leverage the power of individual/population heterogeneity from a collection of 27 transcriptome datasets to establish a generalized multivariate model for TB risk estimation and treatment monitoring. In the process of model development, we first developed a network-based meta-analysis that utilized the collected cohorts containing different clinical/genetic/technical covariates to identify a set of common genes that robustly distinguish ATB from LTBI, HC, OLD and undergoing TB treatment (Tx) across the cohorts. Importantly, the network-based approach was designed to capture not only the most differentiated genes, but also the covariation between genes that elucidated functionally important biological processes that underlie progression to ATB. This approach led to the identification of a biologically relevant and statistically robust gene signature. We then trained two optimized machine-learning (ML) regression models (with full and reduced gene sets) to connect the expression patterns of the gene signature and the dynamic continuum of *Mtb* infection to ATB disease. A systematic validation of the model is presented using 10 independent longitudinal studies to evaluate the potential utility of our model in a real-world setting and 20 additional viral infection studies for testing model specificity to TB. Finally, we establish a probabilistic model to estimate TB disease risk for use as a TB screening tool.

## Results

### A network-based meta-analysis approach identifies a biologically meaningful common gene signature specific to active tuberculosis

First, we aimed to identify a set of common genes consistently differentially expressed in ATB cases relative to other conditions, including HC, LTBI, OLD and Tx, while accounting for cohort/population heterogeneity among the studies. To accomplish this, we developed a network-based meta-analysis approach where we established a gene covariation network to consider genes that were both differentially expressed, and co-varied among multiple studies based on the meta-analysis (**Fig 1A**). We identified a TB-specific common gene set and used it as the training dataset to build the ML predictive model for TB risk estimation and treatment monitoring. Specifically, we collected whole blood transcriptome data from 27 published studies (the discovery dataset) in which each dataset had ATB and at least one other contrasting condition for a given cohort of individuals (**Tables 1 and S1**). For each contrasting condition comparison (e.g., ATB vs. HC) we conducted a differential gene expression analysis within a cohort resulting in a list of differentially expressed genes with corresponding log fold-changes (logFC) (**Fig 1A1**). After repeating the analyses across cohorts, we created an $N \times M$ matrix $X$ with logFC values for $M$ cohorts and $N$ genes (**Fig 1A2**). We further simplified $X$ by converting the matrix values to +1, 0, -1 to account for uncertainty of the data, resulting in a matrix that

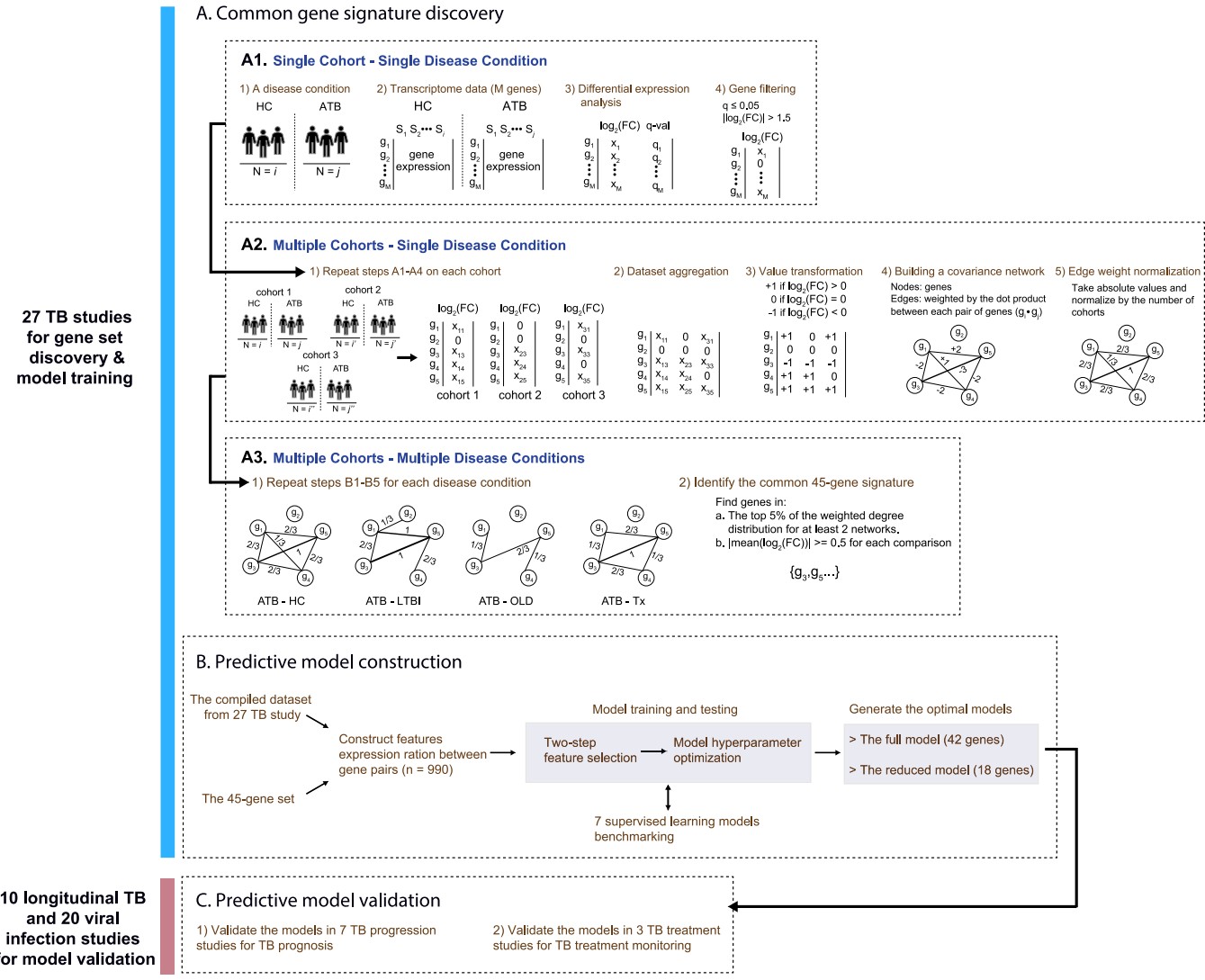

**Fig 1. An end–to–end gene signature model development scheme.** (**A**) The network–based meta–analysis approach. (**A1**) Schematic of differential gene expression calculations for a single cohort given a disease condition. (**A2**) Steps for integrating differential gene expression results for multiple cohorts into a gene covariation network. (**A3**) Four different networks were constructed corresponding to four different disease conditions: ATB v HC, ATB v LTBI, ATB v OLD, ATB v Tx. From these four networks, the top weighted nodes across networks were selected to form the common ATB–specific gene signature. (**B**) ML predictive model training and testing based on the common gene signature. (**C**) Steps for the predictive model validation.

captured differentially expressed genes that were either upregulated or downregulated between ATB and the other conditions (HC, LTBI, OLD or Tx).

To capture co-varying genes across cohorts, we constructed a gene covariation network from the matrix $X$ where each node represented a gene, and computed the edge weight between nodes by taking the dot product between each pair of genes in $X$. We only retained edges in the network with weights $\geq 3$, which we determined to represent significant covariation from permutation tests (**Figs 1A2 and S1**). We then calculated the node centrality for each node, or the weighted degree of each node as the sum of the weights of the edges assigned to the node (**S2 Fig**). We initially observed that the closer to the center (higher weighted degrees) the nodes were placed in their respective networks, the more consistently the genes responded to active TB among various studies (**Fig 2H–2K**). To define a robust gene signature

**Table 1. Cohort lists used for model training and validation.** TB–related datasets and used for training and validation analyses, along with accession IDs, platforms, and which disease comparisons each dataset was used in (see **S1 Table** for full details including which datasets were used in each network, and details for the respiratory viral infection datasets for model validation). Abbreviations: Net/ML = Network model & ML training/testing; Val = ML validation.

| GEO/EMBL-EBI accession | TB states | Age | Country | HC | LTBI | ATB | OLD | Tx | None | Total | Purpose |
|---|---|---|---|---|---|---|---|---|---|---|---|
| GSE19439 | TB disease | > 18 years | UK | 12 | 17 | 13 | | | | 42 | Net/ML |
| GSE19442 | TB disease | > 18 years | South Africa | | 31 | 20 | | | | 51 | Net/ML |
| GSE19444 | TB disease | > 18 years | UK | 12 | 21 | 21 | | | | 54 | Net/ML |
| GSE28623 | TB disease | 16–54 years | The Gambia | 37 | 25 | 46 | | | | 108 | Net/ML |
| GSE29536 | TB disease | 10–88 years | UK | 6 | | 9 | 395 | | | 410 | Net/ML |
| GSE34608 | TB disease | 17–73 years | Germany | 18 | | 8 | 18 | | | 44 | Net/ML |
| GSE37250 | TB disease | > 17 | Malawi | | 167 | 195 | 175 | | | 537 | Net/ML |
| GSE39939 | TB disease | < 15 years | Kenya | | 14 | 79 | 64 | | | 157 | Net/ML |
| GSE39940 | TB disease | < 15 years | South Africa, Malawi | | 54 | 111 | 169 | | | 334 | Net/ML |
| GSE41055 | TB disease | < 15 years | Venezuela | 9 | 9 | 9 | | | | 27 | Net/ML |
| GSE42825 | TB disease | > 18 years | UK | 23 | | 8 | 11 | | | 42 | Net/ML |
| GSE42826 | TB disease | > 18 years | UK | 52 | | 11 | 39 | | | 102 | Net/ML |
| GSE42830 | TB disease | > 18 years | UK | 38 | | 16 | 41 | | | 95 | Net/ML |
| GSE50834 | TB disease | 30–40 years | South Africa | 23 | | 21 | | | | 44 | Net/ML |
| GSE56153 | Treatment | > 15 years | Indonesia | 18 | | 18 | | | 35 | 71 | Net/ML |
| GSE54992 | TB disease / Treatment | 18–68 years | China | 6 | 6 | 27 | | | | 39 | Net/ML |
| GSE62147 | Treatment | 15–79 years | Germany | | | 14 | 12 | | 26 | 52 (28) | Net/ML |
| GSE62525 | TB disease | | Taiwan | 14 | 14 | 14 | | | | 42 | Net/ML |
| GSE69581 | TB disease | > 17 years | South Africa | | 25 | 15 | | | 10 | 50 | Net/ML |
| GSE73408 | TB disease | 12–18 years | US | | 35 | 35 | 39 | | | 109 | Net/ML |
| GSE83456 | TB disease | | UK | 61 | | 92 | 49 | | | 202 | Net/ML |
| GSE84076 | TB disease | > 18 years | Brazil | 12 | 16 | 8 | | | | 36 | Net/ML |
| GSE101705 | TB disease | > 6 years | India | | 16 | 28 | | | | 44 | Net/ML |
| GSE107993 | Infection | 16–84 years | UK | 69 | 69 | | | | | 138 | Net/ML |
| GSE107994 | TB disease | 16–84 years | UK | 50 | 72 | 53 | | | | 175 | Net/ML |
| GSE40553 | Treatment | > 17 years | UK, South Africa | | | | | 131 | | 131 | Net/ML |
| GSE31348, GSE36238 | Treatment | 18–65 years | South Africa | | | | | 153 | | 153 | Net/ML |
| GSE116014 | Infection | 12–18 years | South Africa | | | | | | | 85 | Val |
| GSE79362 | TB progression | 12–18 years | South Africa | | | | | | | 214 | Val |
| GSE94438 | TB progression | 10–60 years | South Africa, Ethiopia, Uganda, The Gambia | | | | | | | 418 | Val |
| GSE107995 | TB progression | 16–84 years | UK | | | | | | | 71 | Val |
| E-MTAB-6845 | TB progression | | UK | | | | | | | 108 | Val |
| GSE112104 | TB progression | > = 18 years | Brazil | | | | | | | 71 | Val |
| GSE157657 | TB progression | 16–84 years | UK | | | | | | | 760 | Val |
| GSE89403 | Treatment | | South Africa | | | | | | | 367 | Val |
| GSE157657 | Treatment | 16–84 years | UK | | | | | | | 760 | Val |
| GSE67589 | Treatment | | South Africa | | | | | | | 57 | Val |
| GSE101702 | | 17–90 years | Australia, Canada, Germany | Influenza | | | | | | Viral | Val |
| GSE103842 | | 0–2 years | USA | RSV | | | | | | Viral | Val |
| GSE111368 | | 18–71 years | UK | Influenza | | | | | | Viral | Val |
| GSE117827 | | 0–11 years | USA | Rhinovirus, RSV, Enterovirus, Coxsackievirus | | | | | | Viral | Val |

*(Continued)*

**Table 1.** (Continued)

| GEO/EMBL-EBI accession | TB states | Age | Country | HC | LTBI | ATB | OLD | Tx | None | Total | Purpose |
|---|---|---|---|---|---|---|---|---|---|---|---|
| GSE17156 | | >18 years | USA, UK | Influenza, Rhinovirus, RSV | | | | | | Viral | Val |
| GSE20346 | | 21–75 years | Australia | Influenza | | | | | | Viral | Val |
| GSE21802 | | 18–65 years | Spain | Influenza | | | | | | Viral | Val |
| GSE25504 | | 0–1 year | UK | Rhinovirus, CMV | | | | | | Viral | Val |
| GSE38900 | | 0–2 years | USA, Finland | RSV, Rhinovirus, Influenza | | | | | | Viral | Val |
| GSE40012 | | 22–75 years | Australia, Hong Kong | Influenza | | | | | | Viral | Val |
| GSE4607 | | 0–10 years | USA | Influenza, Rotavirus, Varicella | | | | | | Viral | Val |
| GSE61754 | | 18–45 years | UK | Influenza | | | | | | Viral | Val |
| GSE61821 | | 5–73 years | Singapore, Thailand, Indonesia, Vietnam | Influenza, Other Viral | | | | | | Viral | Val |
| GSE6269 | | 0–18 years | USA | Influenza | | | | | | Viral | Val |
| GSE66099 | | 0–10 years | USA | Influenza, HSV, Adenovirus, BKV, CMV, HMPV, Parainfluenza | | | | | | Viral | Val |
| GSE67059 | | 0–2 years | USA, Finland, Spain | Rhinovirus | | | | | | Viral | Val |
| GSE68004 | | 0–16 years | USA | Adenovirus | | | | | | Viral | Val |
| GSE68310 | | 18–49 | USA | Influenza, Rhinovirus, Coronavirus, RSV, Enterovirus | | | | | | Viral | Val |
| GSE73072 | | 18–41 years | USA | Influenza, Rhinovirus, RSV | | | | | | Viral | Val |
| GSE77087 | | 0–2 years | USA | RSV | | | | | | Viral | Val |

discriminating ATB from different conditions, we built four covariation networks for each condition comparison and rank ordered genes based on their centrality (**Fig 1A3**). Finally, we identified a set of 45 genes that were in the top 5% of central genes from the weighted degree distribution of the network, were central in at least two of four networks, and had average logFC $\geq$ 0.5 or $\leq$ -0.5 across the studies (**Fig 2A–2E**).

We validated this gene set with a volcano plot showing the average logFC across datasets against the weighted degree of each gene for each network constructed and highlighted the 45 gene set selected (**Fig 2H–2K**). Consistent with our selection strategy, we found that most genes in this gene set were highly differentiated when averaged across cohorts for each disease comparison and heavily weighted (center) in most of the networks. In addition, we plotted the differential expression of the 45 genes across all datasets (**Fig 2L–2O**) and observed that gene patterns of upregulation and downregulation were consistent across cohorts within each disease condition and between disease conditions. Genes upregulated in ATB vs. HC for example were also upregulated in ATB vs. OLD, demonstrating a consistent signal for this gene set across cohorts within and between disease conditions. To validate this pattern for the 45-gene set, we averaged the logFC values for each gene across datasets within each comparison (ATB vs HC, ATB vs LTBI, ATB vs OLD, and ATB vs Tx) and computed the correlation between these averaged expression patterns between each pair of comparisons (e.g. ATB vs HC & ATB vs LTBI) to construct a correlation matrix (**S3 Fig**). We observed that these differential expression patterns correlated strongly across disease conditions with $r$ ranging between 0.61–0.96 and all p-values $\leq$ 1e-05. However, a few genes (such as HP, AEACAM1) showed slightly inconsistent expression patterns in ATB vs. OLD.

We next sought to identify a biological interpretation for this 45-gene set that contained several genes used in prior TB gene signature studies (**S4 Fig**). First, we performed a gene set enrichment analysis and found an enrichment of genes involved in immune function such as interferon gamma (IFN-γ) and alpha/beta (α/β) signaling, IL-6 signaling and Toll-like receptor cascades (**Fig 2G** and **S2 Table**). Next, we mapped all 45 genes in the protein-protein

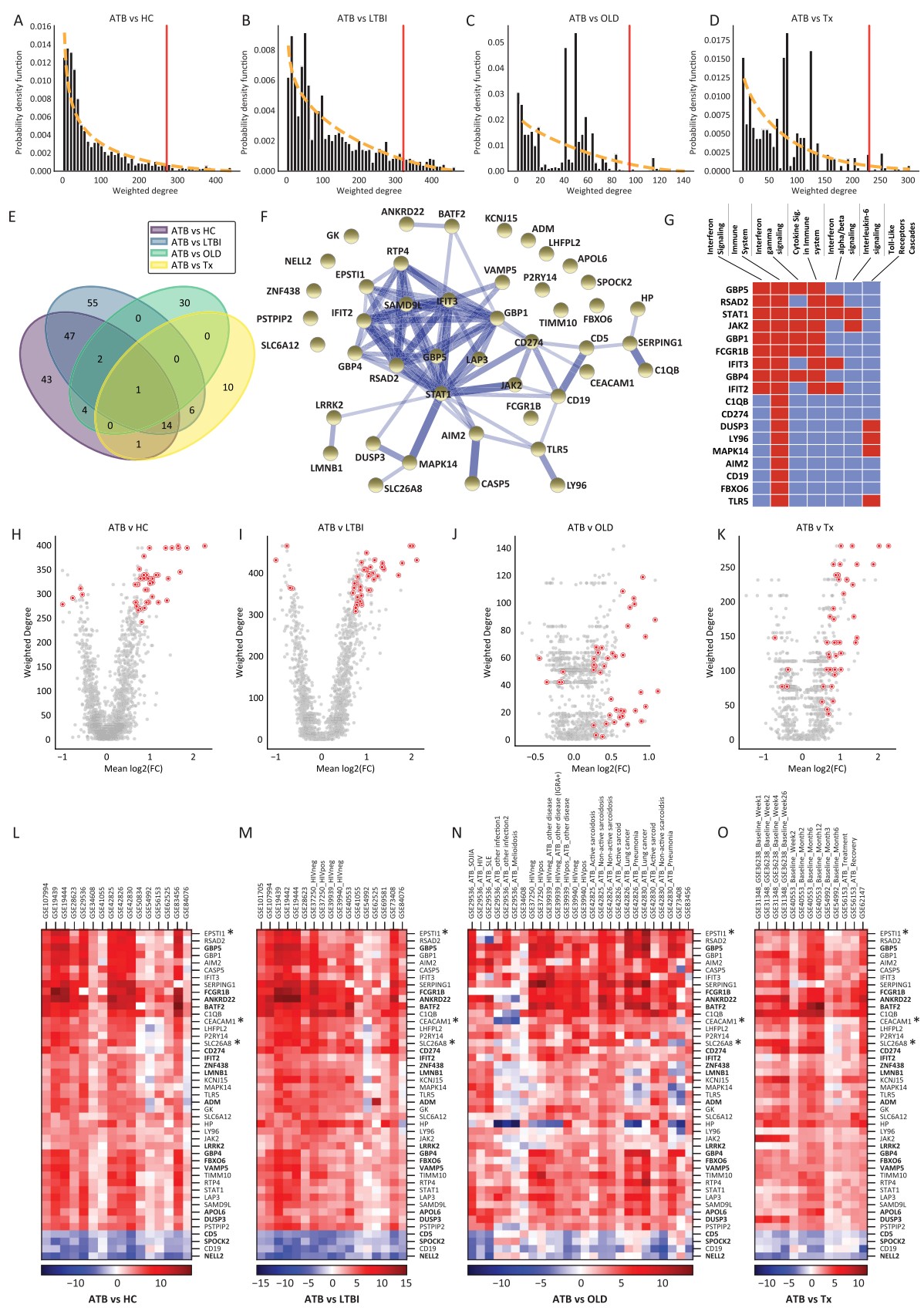

**Fig 2. Network analysis results and gene signature.** (**A–D**) Network degree distribution fitting. The distribution of weighted degrees for each network was fit with a probability density function to help determine top–ranked genes to retain for the candidate gene set (**Methods**). (**E**) Overlap of top 5% of genes by weighted degree in each network. (**F**) Gene signature mapped on protein–protein interaction network. Protein–protein association network of 45 candidate genes constructed with STRING. Width of edge corresponds to edge confidence. Associations correspond to physical interactions or represent proteins that act functionally together. (**G**) A visualization of S2 Table, showcasing which pathways (and corresponding genes) were enriched for our 45–gene set (red: gene is in pathway, blue: gene is not part of pathway). (**H–K**) Volcano plots displaying the mean log2(Fold Change) of gene expression across cohorts against the weighted degree for each network for all genes within each network. Forty–five candidate genes highlighted in orange within each plot. (**L–O**) Heatmaps of log2 (Fold Change) for each of the 45 candidate genes (rows) between disease conditions displayed for each cohort (columns) that were used in constructing each network. The corresponding GSE ID of each cohort was labeled on columns. If a cohort contains multiple clinically defined populations, a specific comparison is also highlighted on the column. Bolded gene names are genes that are also included in the reduced model, genes with an asterisk are not included in either the full or reduced model.

association network using the STRING database [39]. Strikingly, we found 71% (32/45) of the genes were interconnected (**Fig 2F**). The connections between genes may indicate physical interaction between some of the encoded proteins, or that the proteins jointly contribute to a shared function [39]. Proteins IFIT2 and IFIT3 centered in the network are IFN-α/β induced proteins and highly expressed during cell-intrinsic immune response to *Mtb* infection [40]. Further, we observed that STAT1 forms associations with many proteins in the network ($n = 17$); STAT1 is known to be an immunoregulatory factor that regulates both IFN-α/β and IFN-γ -mediated responses [7,41,42]. Taken together, this report using a network-based meta-analysis approach on whole blood transcriptome datasets provided the first systematic evidence to identify a set of 45 genes that commonly appeared in all 27 studies and was enriched for genes actively involved in interferon activation responding to infection.

## Two novel machine-learning (ML) models predict TB disease states

The scope of the ML work was to generate a TB score, a continuous variable, with a fixed range between 0 and 1, representing a dynamic continuum from *Mtb* infection to disease as well as following disease treatment back to cured. We employed multivariate regression to establish the relationship between outcome (TB disease states) and the expression patterns of the ATB-specific gene signature identified from our network-based meta-analysis (**Fig 1B, Materials and Methods**). To capture heterogenous responses of individuals in different TB states among the cohorts, we pooled together the whole discovery dataset (27 studies) (**Tables 1 and S1**), and the ML model regressed to two states: 1 (ATB) and 0 (HC, LTBI, Tx or OLD). The ML framework began with feature selection where the features fed into ML models were derived from an expression ratio between any pair of the common genes (**Materials and Methods**). Feature selection underwent two steps: univariate and multivariate feature selection. Combining these two steps allowed removal of irrelevant or redundant features from the original 990 gene-paired features down to 41 features which included 42/45 genes (**S4 Table**) and mitigated the risk of model overfitting while improving predictive performance. We then assessed the best models among 7 ML regression algorithms using the discovery dataset based on 5-repeated 5-fold nested CV (cross-validation) framework. The random forest (RF) model consistently outperformed the other 6 ML models based on the performance metrics (average outer cross-validation $R^2 = 0.51$, MSE = 0.11, AUROC = 0.91), while multilayer perceptron model, an artificial neural network approach, was second-best (average outer cross-validation $R^2 = 0.48$, MSE = 0.11, AUROC = 0.90) (**S3 Table** and **S5 Fig**). We generated a final RF model trained using the full discovery dataset; we refer to this model as the full model, its parameters are listed in **S4 Table**.

Given that blood transcriptomic signatures hold great promise for development of screening and triage tests but they impose a limit on the number of genes they can probe [30,32,38],

we sought to construct a model with fewer features but without reducing predictive performance. We further adapted a stability selection approach to perform a sensitivity analysis and down-selected the most robust features while preserving predictive performance (S6 Fig). Using a forward stepwise selection within a 5-fold nested CV framework, we evaluated the RF algorithm by adding one feature at a time starting from the top ranked features. This final model with 12 features (paired genes) was determined with slightly reduced performance (average outer cross-validation AUROC = 0.85) (S7 Fig) and refer to this model as the reduced model.

As a result, we generated two models, labeled as the full and reduced model (S4 Table). Importantly, among the 18 genes selected by the reduced model, 8/18 genes (ANKRD22, APOL6, BATF2, DUSP3, FCGR1B, GBP4, GBP5, and VAMP5) were repeatedly identified by the best performing gene signature models for incipient TB diagnosis reported previously [43] (S4 Fig), in which DUSP3 and GBP5 were included in the Sweeney 3 gene signature and have been tested in a PoC setting for TB diagnosis [5,38,44]. Seven additional genes (ADM, CD274, FBOX6, LMNB1, IFIT2, NELL2, and ZNF438) have been identified as part of several gene signatures associated with active TB [7,28,45,46]. For the remaining 3 new genes (CD5, LRRK2 and SPOCK2), CD5-expressing regulatory B cells were found to negatively regulate Th17 and to associate with active TB [47]. SPOCK2 expression was shown to be associated with lung injury induced by viral infection [48] and the mouse LRRK2 knockout model suggested that LRRK2 regulates innate immune responses and lung inflammation during *Mtb* infection [49]. Taken together, we demonstrate that our gene signature discovery approach is robust enough to produce comparable results to other models, and to expand on previous models with better performance.

## Validation of the gene signature models in TB disease risk prediction

To systematically validate the models, we collected 10 independent longitudinal studies (7 TB progression and 3 TB treatment cohorts) from multiple countries and different target populations (Fig 1C) (Tables 1 and S1). We first pooled all 7 TB progression datasets [17–19,27,28,31] and stratified the data using different time intervals to disease and LTBI without progression. TB scores generated from the full model showed a graded increase along time intervals beginning >2 years to disease, and the score distributions from all progression groups differed from the no progression group (Fig 3A). Similar observations were found in the reduced model (S8 Fig). Furthermore, one of the progression studies (the Leicester household contact study [31]) categorized subjects into incipient, subclinical, or clinical TB, according to their clinical phenotypes at the time of sampling. The full model was able to statistically separate the subjects' scores between clinically defined subgroups and healthy controls as well as in-between clinically defined subgroups (with the exception of subclinical TB & clinical TB) (Fig 3B), suggesting that the TB scores derived by the RF model recapitulated host responses to clinical TB pathogenesis. With the same pooled dataset, we compared TB scores generated by four previously published models. These models demonstrated accurate performance in short-term TB risk estimation [3,5,18,27]. We observed similar trends between the scores and time intervals to disease among the models, except for the Sweeney 3 signature; this model may be sensitive to different cohort populations, resulting in the bimodal distributions we observed in several groups (S9 Fig). For most of the published models the score distributions in the groups distant to disease progression (> 12 months) were less discriminant to the no progression group, which poses a challenge on incipient TB diagnosis at early timepoints (S9 Fig).

We next examined prognostic performance of the models by computing AUROCs, stratified by multiple time intervals to disease (<3, <6, <12, <18, <24, <30 months). Alternatively,

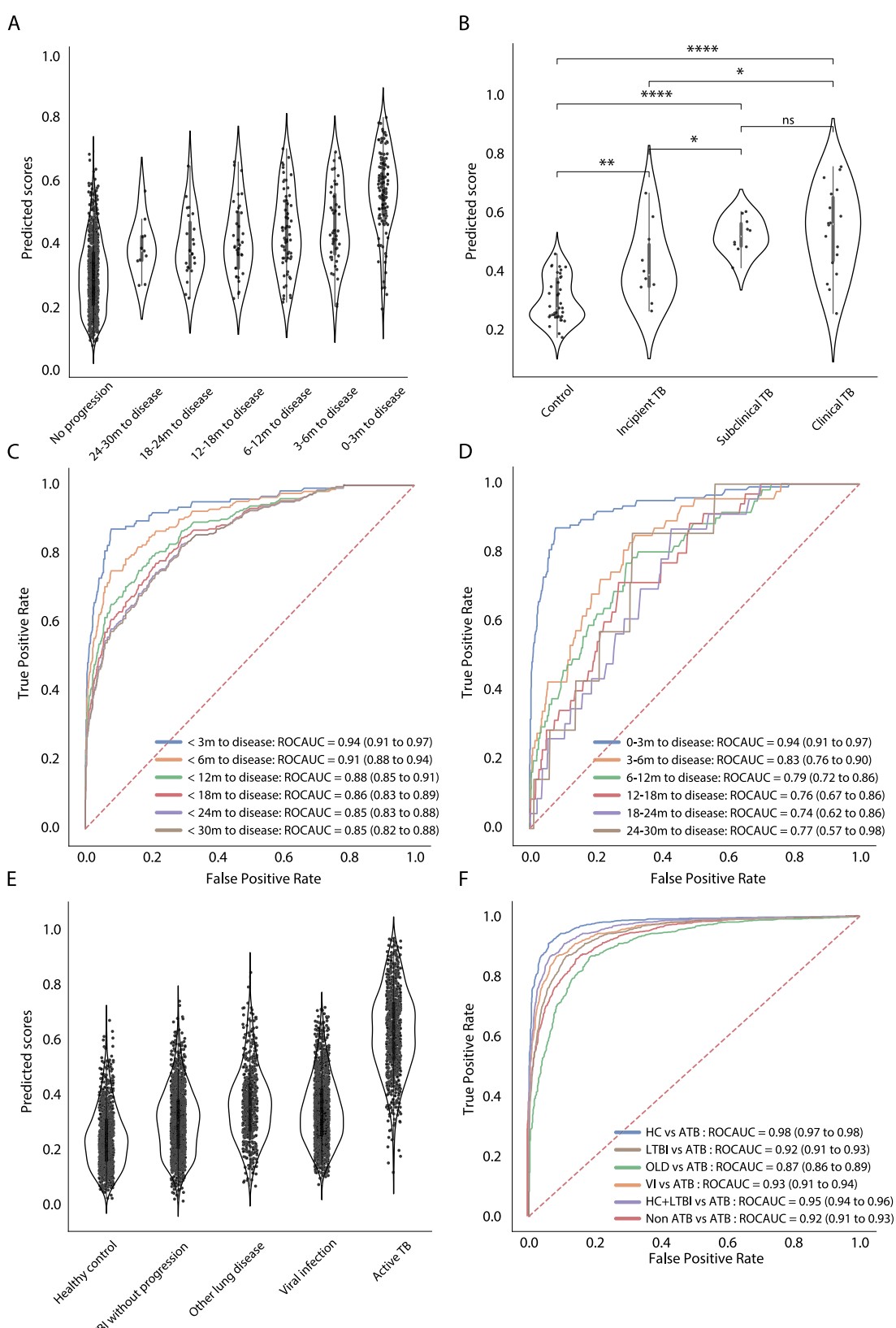

**Fig 3. Systematic model validation using TB progression cohorts.** (**A**) The distributions of TB scores generated by the full model, stratified by categorical interval to disease (datapoints $n$ = 1281), and (**B**) clinically defined TB states (datapoints $n$ = 137) are shown in the violin plot. Individual datapoints are plotted as a point in the violin plot. P–values were calculated using the Mann–Whitney U test with Bonferroni correction (ns, $p < 1$; *, $p < 0.05$; **, $p < 0.01$; ***, $p < 0.001$; ****, $p < 0.0001$). (**C–D**) Receiver operating characteristic curves depict diagnostic performance of the model for incipient TB, (**C**) stratified by time intervals to disease ($< 3$, $< 6$, $<12$, $<18$, $< 24$, $< 30$ months) and (**D**) mutually exclusive time intervals to disease (0–3, 3–6, 6–12, 12–18, 18–24, 24–30 months). (**E**) The distributions of TB scores generated by the full model, stratified by different TB disease stages (HC, LTBI, and ATB), other lung disease (OLD) and viral infection (VI), are visualized in the violin. (**F**) Area under the curve and 95% confidence intervals for each interval to disease are also shown. Diagnostic performance of the model in differentiating between ATB versus HC, LTBI, OLD, and/or viral infection (VI) using a pooled dataset of all 57 collected cohort studies (**S1 Table**) (datapoints $n$ = 6290) are showed in ROC curves.

we examined mutually exclusive time intervals to disease (0–3, 3–6, 6–12, 12–18, 18–24, 24–30 months) as a sensitivity analysis. For the full model, AUROC for identification of incipient TB within 3 months was able to achieve 0.94 (95% CI 0.91–0.97) and declined along with increasing interval to disease. Despite this, the AUROC for the group over a 2-year period ($< 30$ months) was still 0.85 (95% CI 0.82–0.88) (**Fig 3C** and **Table 2**). Strikingly, we observed similar AUROCs in the reduced model even though this model incorporated fewer features (**S8 Fig** and **Table 2**). Both models outperformed the previously published models in discriminating between progressors and non-progressors in all time periods, except for the RISK6 signature which provided a comparable but moderately reduced AUROC (**S9 Fig** and **Table 2**). Furthermore, we investigated prognostic performance in each mutually exclusive time-period across the models. AUROC values for the full model ranged from 0.94 (0–3 months, 95% CI 0.91 to 0.97) to 0.74 (18–24 months, 95% CI 0.62 to 0.86); values for the reduced model ranged from 0.94 (0–3 months, 95% CI 0.91 to 0.97) to 0.73 (12–18 months, 95% CI 0.63 to 0.82). We observed no statistical difference between the two models (**Table 3**). Compared to the previously published models, both of our models had better discrimination between progressors and non-progressors in all mutually exclusive time periods except RISK6, which performed slightly better than our full model 6–12 months prior to disease (**Table 3**). Remarkably, when assessing prognostic performance, our models outperformed other models, particularly for the groups distant to disease (12–18, 18–24, and 24–30 months) with AUROCs ranging from 0.74 to 0.77 for the full model and 0.73–0.89 for the reduced model (compared to range 0.67–0.70 for RISK6) (**Table 3**).

Next, we tested the sensitivity and specificity of each model using cut-offs identified by the maximal Youden Index, which is based on the best tradeoff between sensitivity and specificity from each ROC of the time interval. Both of our models met or approximated the minimum criteria ($>75$% sensitivity and $>75$% specificity) in the WHO target product profile (TPP) for prediction of progression to TB disease [50] over a 30-month period (full model sensitivity 74.2% and specificity 78.3%, reduced model sensitivity 77.3% and specificity 74.9%). RISK 6

**Table 2. Prognostic performance of models stratified by inclusive time intervals.** Prognostic performance (AUROC with 95% confidence intervals) of the models developed in this report and published previously on the combined validation datasets for identification of incipient TB within a 2.5–year period, stratified by inclusive time interval to disease.

| Models | < 3m to disease | < 6m to disease | < 12m to disease | < 18m to disease | < 24m to disease | < 30m to disease |
|---|---|---|---|---|---|---|
| Full model | 0.94 (0.91 to 0.97) | 0.91 (0.88 to 0.94) | 0.88 (0.85 to 0.91) | 0.86 (0.83 to 0.89) | 0.85 (0.83 to 0.88) | 0.85 (0.82 to 0.88) |
| Reduced model | 0.94 (0.91 to 0.97) | 0.89 (0.85 to 0.92) | 0.87 (0.84 to 0.90) | 0.85 (0.82 to 0.88) | 0.85 (0.82 to 0.88) | 0.85 (0.82 to 0.88) |
| Sweeney 3 | 0.68 (0.63 to 0.74) | 0.62 (0.57 to 0.66) | 0.62 (0.58 to 0.66) | 0.60 (0.56 to 0.64) | 0.59 (0.55 to 0.63) | 0.59 (0.55 to 0.63) |
| RISK 6 | 0.92 (0.89 to 0.95) | 0.87 (0.84 to 0.91) | 0.85 (0.82 to 0.89) | 0.83 (0.80 to 0.86) | 0.82 (0.79 to 0.85) | 0.82 (0.79 to 0.85) |
| BATF2 | 0.68 (0.62 to 0.73) | 0.65 (0.60 to 0.69) | 0.62 (0.58 to 0.66) | 0.60 (0.56 to 0.64) | 0.59 (0.55 to 0.63) | 0.58 (0.55 to 0.62) |
| Suliman 4 | 0.90 (0.86 to 0.93) | 0.84 (0.81 to 0.88) | 0.80 (0.77 to 0.84) | 0.77 (0.74 to 0.81) | 0.76 (0.73 to 0.79) | 0.76 (0.72 to 0.79) |

**Table 3. Prognostic performance of models stratified by exclusive time intervals.** Prognostic performance (AUROC with 95% confidence intervals) of the models developed in this report and published previously on the combined validation datasets for identification of incipient TB within a 2.5–year period, stratified by mutually exclusive time interval to disease.

| Models | 0-3m to disease | 3-6m to disease | 6-12m to disease | 12-18m to disease | 18-24m to disease | 24-30m to disease |
|---|---|---|---|---|---|---|
| Full model | 0.94 (0.91 to 0.97) | 0.83 (0.76 to 0.90) | 0.79 (0.72 to 0.86) | 0.76 (0.67 to 0.86) | 0.74 (0.62 to 0.86) | 0.77 (0.57 to 0.98) |
| Reduced model | 0.94 (0.91 to 0.97) | 0.75 (0.67 to 0.83) | 0.84 (0.77 to 0.90) | 0.73 (0.63 to 0.82) | 0.75 (0.64 to 0.87) | 0.89 (0.73 to 1.00) |
| Sweeney 3 | 0.68 (0.63 to 0.74) | 0.44 (0.35 to 0.52) | 0.63 (0.55 to 0.71) | 0.49 (0.39 to 0.58) | 0.47 (0.35 to 0.59) | 0.58 (0.36 to 0.80) |
| RISK 6 | 0.92 (0.89 to 0.95) | 0.75 (0.67 to 0.83) | 0.80 (0.73 to 0.87) | 0.69 (0.60 to 0.79) | 0.67 (0.55 to 0.79) | 0.70 (0.48 to 0.92) |
| BATF2 | 0.68 (0.62 to 0.73) | 0.56 (0.47 to 0.64) | 0.54 (0.47 to 0.62) | 0.50 (0.40 to 0.59) | 0.46 (0.34 to 0.58) | 0.30 (0.14 to 0.47) |
| Suliman 4 | 0.90 (0.86 to 0.93) | 0.70 (0.61 to 0.79) | 0.69 (0.61 to 0.76) | 0.57 (0.47 to 0.67) | 0.62 (0.49 to 0.74) | 0.60 (0.38 to 0.83) |

and Suliman 4 only met the criteria over 0–12 months and 0–6 months, respectively. Sweeney 3 and BATF2 did not meet the criteria in any of the time periods (**S5 Table**). Based on a pretest probability of 2% with Youden Index cutoff, we estimated the PPV (positive predictive value) and NPV (negative predictive value) of each model. Both of our models achieved a PPV ranging from 16.8% to 17.7% for a 0–3 month and marginally surpassed the WHO target PPV of > 5.8% over 2 years (full model PPV 6.6% and reduced model PPV 5.9%). None of the other models except RISK 6 met the WHO PPV criteria over 0–2 years (RISK 6 PPV 5.9%) (**S5 Table**). We then used a pre-specified cutoff of 2 standard deviations (SDs) above the mean of the non-progressing group to prioritize PPV and specificity, which was proposed in previous reports [27,43]. At this threshold, the PPVs estimated in our models significantly improved to 27.9% and 30% over 0–3 months, and 19.5% and 20.3% over 0–24 months for the full and reduced model, respectively, which substantially surpassed the WHO target PPV over 0–2 years. We observed similar improvements in the PPV for the RISK 6 and Suliman 4 models (**S6 Table**). However, overall sensitivities using this cutoff were significantly reduced compared to the one using Youden Index although specificities remained high (> 96% in all models). Sensitivities over 0–3 months were 66.7% for the full model and 69% for the reduced model. Over 0–30 months, sensitivities were 41.1% for both of our models. Many cases at distant time points to disease progression (>12 months) could not be detected by using this cutoff.

## Gene signature models discriminate active TB from non-active TB

Two published gene signature models (Sweeney 3 and RISK 6) demonstrated their utility as a screening or triage test for TB diagnosis in multicenter prospective studies [32,38]. Here we examined diagnostic performance of our models to discriminate active from non-active TB subjects in comparison with the published models using the pooled datasets of all 37 studies. Using this pooled dataset allowed us to recapitulate various confounding variables to mimic a real-world setting. After pooling the datasets, subjects were stratified by different TB states, and the scores for every subject were generated by individual models. Additionally, as Mulenga et al. highlighted, respiratory viral infections induce interferon-stimulated genes, which can be a potential confounding variable for gene signature predictions [37]. We therefore also included an additional 20 datasets containing 15 different respiratory viral infections (**S1 Table**). Note, since our RF models were generated by using part of these pooled datasets (discovery dataset), the assessment of our model performance in this section was subjected to 5-fold nested CV to ensure a blinded model validation. The full model demonstrated score distributions able to distinguish between the groups and provided a strong discrimination between ATB vs. HC, LTBI or OLD (**Fig 3E and 3F**). Additionally, the full model was able to discriminate viral infection (VI) from ATB (AUROC 0.93, 95% CI 0.91–0.94), the overall

AUROC between ATB and non-ATB (HC, LTBI, OLD and VI) achieved 0.92 (95% CI 0.91–0.93). As expected, the reduced model using fewer features had reduced diagnostic performance but maintained an AUROC at 0.87 (95% CI 0.86–0.89) when separating ATB from non-ATB (S10 Fig). Its reduced performance was primarily attributed to insufficient differentiation between ATB and OLD (AUROC 0.75, 95% CI 0.73–0.77) and between ATB and VI (AUROC 0.83, 95% CI 0.81–0.85) (S10 Fig). In contrast, the pooled TB scores generated by the published models exhibited multimodal or dispersed distributions in multiple groups (S11 Fig). This resulted in less overall discrimination between ATB and non-ATB (Sweeney 3 AUROC 0.81, 95% CI 0.79–0.83; RISK 6 AUROC 0.75, 95% CI 0.73–0.77; BATF2 AUROC 0.70, 95% CI 0.68–0.72; Suliman 4 AUROC 0.79, 95% CI 0.75–0.82) (S11 Fig). The results suggest that published models assessing the expression of fewer genes, especially the BATF2 model testing a single gene, may be more sensitive to confounding variables.

## A probabilistic model to assess TB risk

Identifying a positive case from a screening or triage test for disease diagnosis or prognosis usually requires a pre-defined cutoff, which needs to be validated by post-hoc analyses from independent studies. Given heterogeneous blood transcriptomic responses to TB progression due to confounding variables as previously described, this pre-defined cutoff varies in different settings. Similar evidence has been discussed in other reports [3,16,51]. In addition to using the cutoff approach for test readout, here we also propose a new probabilistic model to estimate TB risk using a cutoff-free method while considering individual and population heterogeneity in the model. To build the probability model, we used our full model to compute all TB scores and stratified the datapoints into different TB states (Fig 4A). The score distributions increased according to TB state, beginning with the healthy control and latent infection without progression groups, followed by incipient TB groups ranging from distant to proximal to disease. The active TB group had the highest scores. Using this stratified score data, we generated a probability model to estimate TB risk as a function of TB scores ranging between 0 to 1. The model composed six best-fitted probability functions, each of which estimated the probability associated with the observed outcome (healthy control, latent infection without progression, < 24m, < 12m, < 3m to disease, and active TB), for any given TB score (Fig 4B). For example, a subject with a TB score = 0.2 is estimated to be 82% in LTBI, 58% in HC and only <5% in either incipient or active TB. In contrast, a subject with a TB score = 0.6 is estimated to be 78% in < 24m, 72% in < 12m, 58% in < 3m to disease, 42% in active TB, and only < 5% in either HC or LTBI. As the score increases and approaches 0.8, the probabilities for incipient TB < 3m to disease dramatically increase and approach 100%, as is the case with active TB in which the distribution is shifted further to the right. This probabilistic model allows us to generate an additional readout for TB risk stratification or TB screening while taking into account uncertainty in the data due to biological or technical variation.

## Validation of reduced gene signature model in treatment monitoring and treatment outcome prediction

We have shown that these whole blood gene signatures serve as a correlate of immune status responding to *Mtb* bacterial load during TB progression. Accumulated evidence has also demonstrated the utility of gene signatures in treatment monitoring along with *Mtb* elimination [3,16,30–35]. Here we interrogated whether our gene signature model, in particular the reduced model, could be used to infer treatment responses and predict clinical outcomes (cure, failure, and recurrence). We used two independent treatment studies which provided

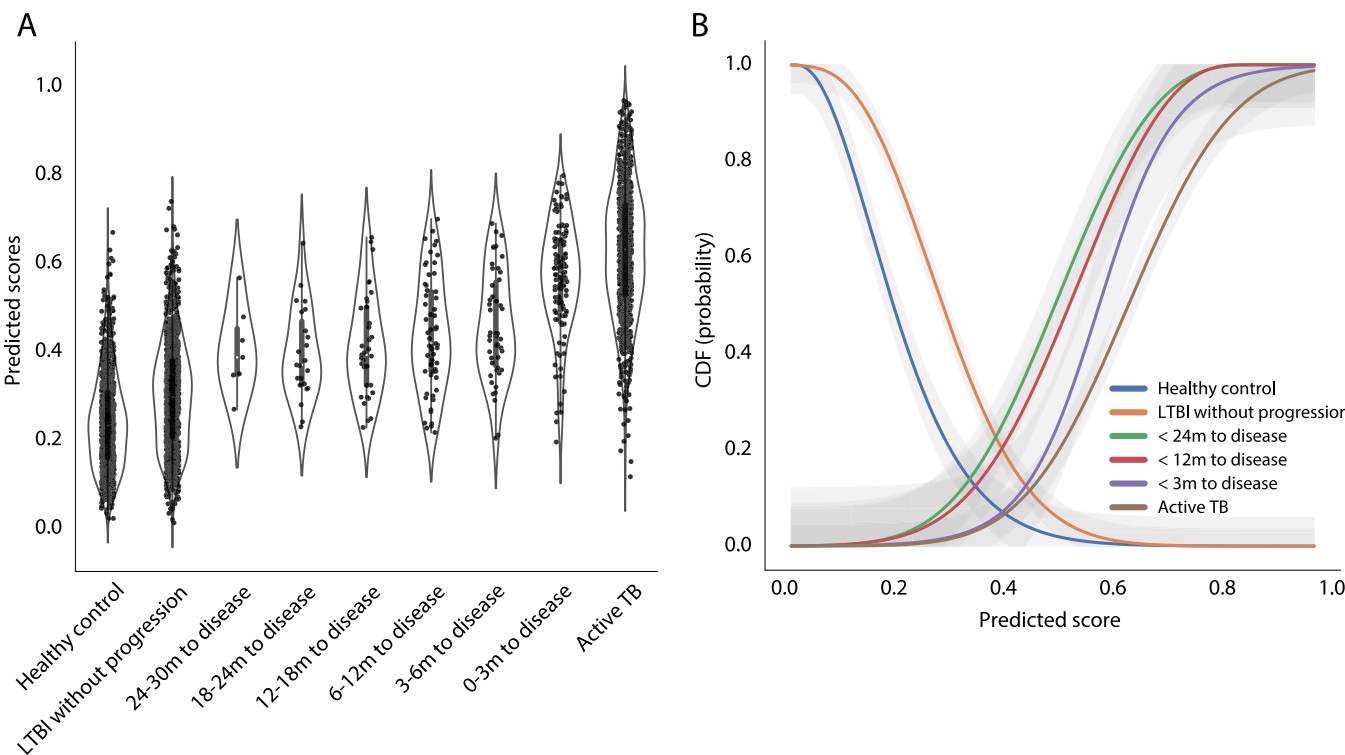

**Fig 4. Building a probabilistic TB risk model.** (**A**) The distributions of the full model–generated TB scores, stratified by categorical interval to disease, and active TB, are shown in the violin plot, based on the pooled dataset collected from all the 37 studies. Individual datapoints of the dataset are plotted as a point in the violin plot (datapoints $n = 3869$). (**B**) The 6 probability functions (cumulative distribution function) that depict the probabilities of the outcomes (HC, LTBI, < 24m, < 12m, < 3m to disease, and active TB) as a function of TB scores were showed in lines with different colors. The shaded area represented the 95% confidence interval area.

clinical characterization with publicly available transcriptome and bioassay data to assess the consistency of our model prediction with the study results [31,35,52].

The Catalysis treatment response dataset contained 96 HIV-uninfected adults with newly diagnosed pulmonary TB undergoing standard 6-month treatment [52]. Of the 96 participants, 8 did not achieve cure and failed treatment when the sputum culture remained positive at month 6, and 12 of 88 cured participants had recurrent TB within 2 years after treatment completion. We first examined the dynamics of TB scores generated by our model in response to treatment. Since the reduced RF model's performance in the prognostic assessment is non-inferior to the full model (**Tables 2 and 3**), and holds greater potential for clinical development, we used only the reduced model for the validation in treatment monitoring. The predicted scores significantly declined during the first week after treatment initiation and continuously decreased with treatment duration. At the end of treatment (EOT) (Day 168), the scores from most of the participants were at a similar level as healthy controls (**Fig 5A**). The model was able to discriminate between healthy controls and before treatment (Day 0) (AUC 0.92, 95% CI 0.87–0.97), 1 week (AUC 0.76, 95% CI 0.67–0.85), and 4 weeks after treatment initiation (AUC 0.72, 95% CI 0.63–0.81). Poor discrimination between healthy controls and EOT was observed which agreed with the clinical outcome where most of the participants (88 out of 96) were cured at EOT (**Fig 5B**).

We further stratified the participants by time to sputum culture conversion to negative (negativity at day 28, 56, 84 and 168, and no conversion at day 168 [treatment failed]). The participants who failed treatment retained high TB scores throughout the course of treatment,

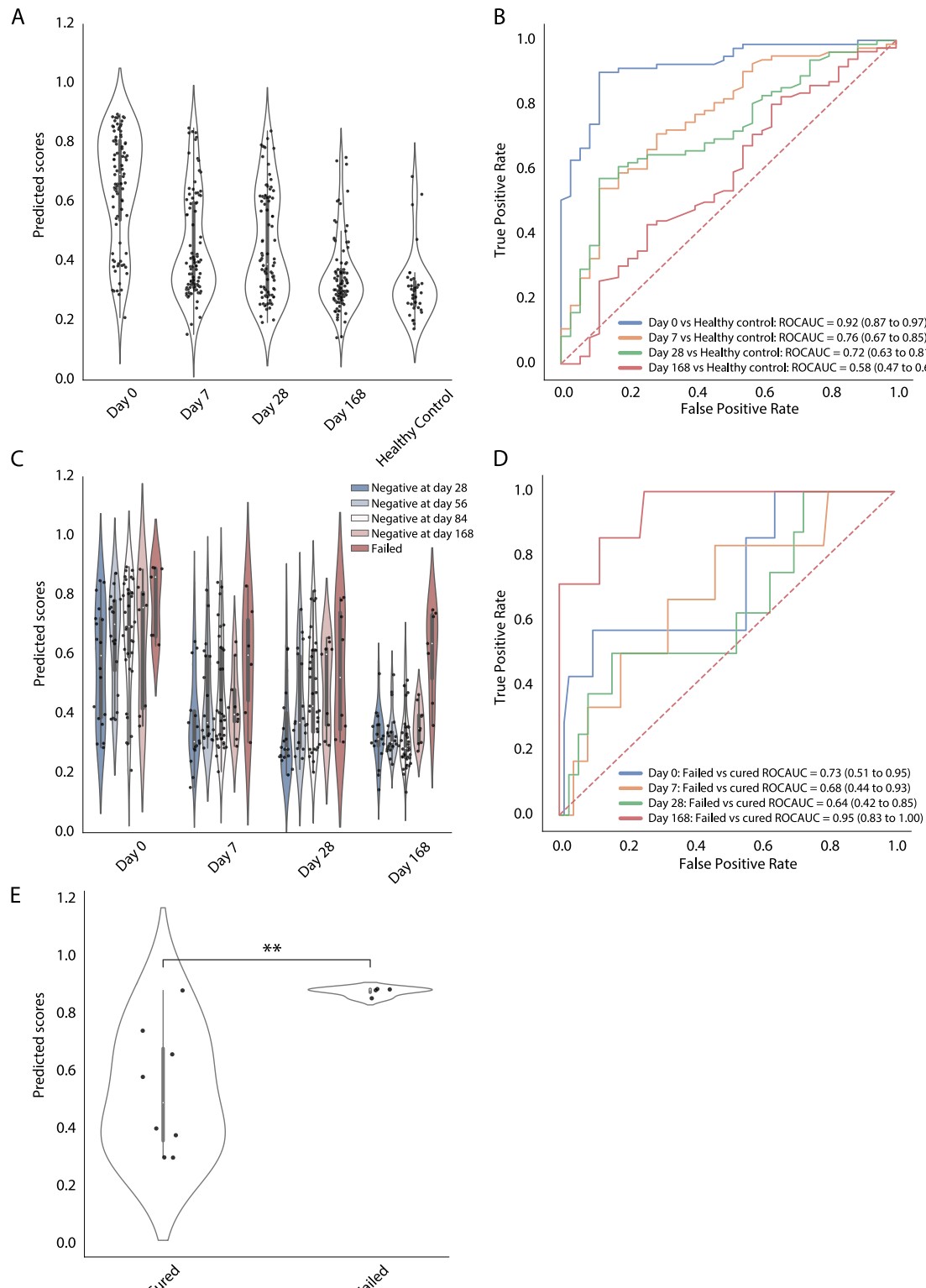

**Fig 5. Model validation using the Catalysis TB treatment study.** (**A**) Comparison of TB scores generated by the reduced model at Day 0, 7, 28, and 168 after treatment initiation and from healthy controls are shown in a violin plot. Individual datapoints are plotted as a point in the violin plot (datapoints $n = 395$). (**B**) Receiver operating characteristic curves depict model discrimination between healthy controls and the patients from different time points after treatment initiation. Area under the curve and 95% confidence intervals for each interval to disease are also shown. (**C**) At each timepoint the TB scores were stratified by the time of

sputum culture conversion to negative (negativity at day 28, 56, 84 and 168, and no conversion at day 168 [failed]) and plotted in the violin plot (datapoints $n$ = 360). (**D**) ROC curves, stratified by different timepoints after treatment initiation, depict predictive performance of the models for discrimination between the patients with bacteriological cure and those with treatment failure at EOT. (**E**) Comparison of TB scores against treatment outcome from poorly compliant subjects at day 0. P–values were calculated using the Mann–Whitney U test (**, $p < 0.01$).

which were statistically different from cured groups. In contrast, at 7 days after treatment initiation the TB scores from the group with the earliest culture conversion to negative (day 28) had approached a similar level as at EOT (**Fig 5C** and S**12**). Importantly, the reduced model statistically discriminated between participants with bacteriological cure and those with treatment failure not only at EOT (AUC 0.95, 95% CI 0.83–1.00) but at baseline (AUC 0.73, 95% CI 0.51–0.95) (**Fig 5D**). RISK 6 and BATF2 also provided a comparable result in treatment outcome prediction (**S14A Fig**). Strikingly, when we compared TB scores at baseline from poorly adherent subjects against treatment outcome (patients who missed more than 20% of treatment were stratified in the group of poor adherence [52]), the subjects with treatment failure showed statistically higher levels of TB scores than those who were cured ($p < 0.01$) (**Fig 5E**).

While treatment failure was statistically associated with poor adherence in this study (p = 0.00032, ANOVA test), we demonstrate that TB scores at baseline serve as another confounding factor that impacts treatment outcome. Moreover, with a portion of cured participants in this study developing recurrent disease within 2 years after treatment completion, we investigated whether the gene signature could differentiate participants with and without recurrence before EOT. The model did not provide enough discriminative evidence to separate the two groups at any timepoint during the course of treatment (**S13 Fig**) and neither did the published models (**S14B Fig**). This could be because relapses were not distinguished from reinfections for these recurrent cases, the models should be retested on validated relapses. Moreover, we investigated whether our model contained properties linked to lung inflammation based on [18F] FDG PET-CT imaging responses in this study, which were reported in RISK 6 and Sweeney 3 [3,44]. The scores generated from our model statistically correlated with lung lesion activity as measured by total glycolytic ratio activity (TGRA) at three time points (Day 0 Spearman $r$ = 0.6, $p < 0.0001$, Day 28 Spearman $r$ = 0.66, $p < 0.0001$, and Day 168 Spearman $r$ = 0.31, $p = 0.0022$) (**S15C–S15E Fig**). Additionally, the TB scores correlated with both MGIT culture time to positivity (Spearman $r$ = -0.62, $p < 0.0001$) and Xpert Ct values (Spearman $r$ = -0.52, $p < 0.0001$); both which were measured before treatment (**S15A and S15B Fig**). Further, the scores at baseline were significantly higher in the participants with radiologically persistent lung inflammation at EOT than those with radiologically cleared lung inflammation (persistent or cleared lung inflammation was defined by the cutoff of TGRA = 400 from a previous report [52]) (**S15F Fig**). Taken together, these findings suggest that the scores from our model are associated to both metabolic activity of TB lung lesion and active *Mtb* infection, both of which contribute to EOT outcome prediction.

We further sought to validate our reduced model on data from another independent treatment study; the Leicester treatment cohort is composed of 74 participants with pulmonary TB. Of these participants, 16 received a 6-month standard regimen treatment (clinical cure < 200 days), 39 received extended treatment (requiring >200 days of standard regimen treatment due to clinical suspicion of TB at EOT), 7 were defined as difficult TB cases (requiring extended treatment due to treatment intolerance and/or adherence issues), 4 were TB drug resistance patients, and 8 were infected with a chronic local outbreak TB strain [31]. Consistent with the Catalysis analysis, we observed declining TB scores after treatment initiation. However, the scores did not reach the same level as healthy control until > 1 year after

treatment initiation, which might be due to 39/74 participants requiring extended treatment (**Fig 6A**). The model consistently discriminated between healthy controls and participants in treatment until month 4. The performance of the model began to wane during month 4–6 (AUC 0.70, 95% CI 0.61–0.79), and month 7–12 (AUC 0.71, 95% CI 0.63–0.80), at which point a portion of the patients had been cured based on clinical assessment (**Fig 6B**).

The average TB score was highest at baseline when compared to the different time intervals, however the score distribution was scattered with a long tail (**Fig 6A**). Strikingly, when we stratified the patients based on smear positive and negative, we found that smear positive patients were statistically more likely to have higher TB scores ($p < 0.0001$). Most of the samples in the long tail of the score distribution were from smear negative patients (**Fig 6C**). Furthermore, the scores from patients requiring extended treatment were consistently higher than those on standard treatment at week 3–4 ($p > 0.05$), month 2–3 ($p < 0.01$) and month 4–6 ($p < 0.0001$) (**Fig 6D–6F**). A previous report identified that smear positive patients mostly fell within the extended treatment patient group [31]. Our model also showed a consistent prediction where patients requiring extended treatment displayed higher scores throughout the course of treatment (**Fig 6G**). These observations suggest that the duration of treatment required in individual patients could be predicted at early timepoints. Indeed, TB scores at month 2–3 were a stronger predictor of treatment duration and significantly discriminated between the patients requiring standard treatment versus those requiring extended treatment after a 6-month treatment (**Fig 6H**). In summary, model validation in two independent treatment studies suggests that our model holds potential to monitor TB treatment success and to predict the duration of treatment required for cure at the end of treatment in TB patients.

## Discussion

Here we develop a network-based meta-analysis with ML modeling to generate a common 45-gene signature specific to active TB disease and to systematically train/validate a generalized predictive model using 57 studies with 6290 data points in total. When applying machine-learning models to omics data, an appropriate feature selection is an effective strategy to reduce feature dimension and redundancy and can alleviate the risk of multicollinearity and model overfitting. We considered several mathematical approaches to perform the feature selection, including lasso regression. However, these methods mathematically consider the dependency between individual genes and the outcome, and do not necessarily account for biological co-variation between the genes across the cohorts. Given that the datasets we collected for our analyses are from diverse TB cohorts, we prioritized leveraging the power of cohort and population heterogeneity to build a generalizable predictive model. To this end, we developed the network modeling approach to capture not only the most differentiated genes relevant to the outcome, but also to incorporate the covariation between genes that elucidated functionally important biological processes that underline TB progression. By leveraging heterogeneity among the included cohorts and implementing rigorous model training and its hyperparameter optimization, we demonstrate robust performance of the model in both short-term and long-term TB risk estimation and treatment monitoring that provides a complementary approach to the current models, most of which offer good performance in ATB diagnosis and/or short-term TB risk estimation. Importantly, our model also shows robust discrimination between ATB and different respiratory viral infections, demonstrating its predictive specificity to ATB. None of the previous models have yet been systematically validated in this aspect.

As many TB gene signatures have been reported previously, we collected 30 previously published gene signatures (**S4 Fig**) [2,43], and found that a large portion of the genes (563 out of

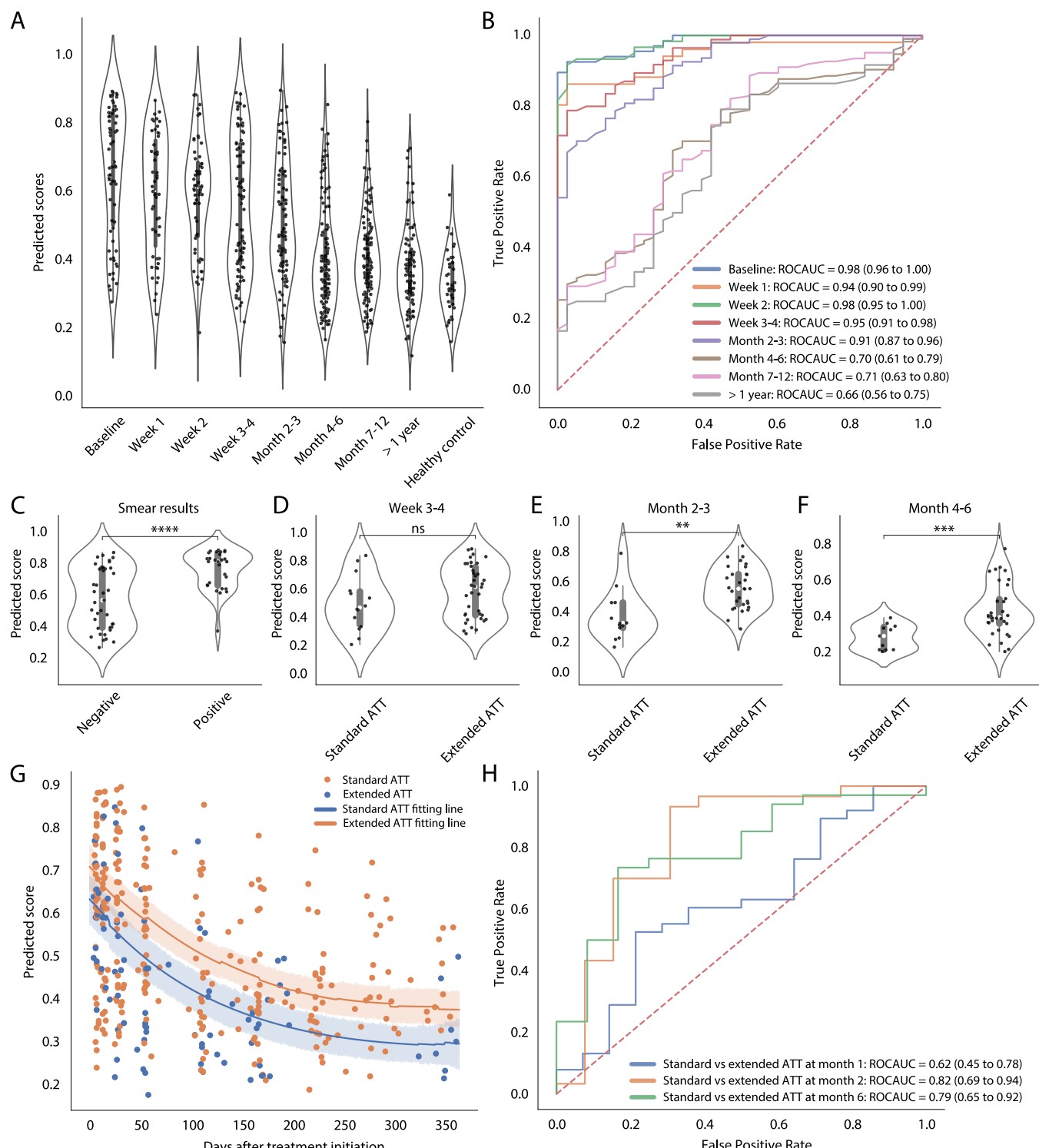

**Fig 6. Model validation using the Leicester TB treatment study.** (**A**) The distributions of the reduced model–generated TB scores, stratified by different time intervals after treatment initiation and healthy control, are shown in the violin plot. Individual datapoints are plotted as a point in the violin plot (datapoints $n = 728$). (**B**) Receiver operating characteristic curves depict model discrimination between healthy controls and the patients from different time intervals after treatment initiation. Area under the curve and 95% confidence intervals for each interval to disease are also shown. Comparison of TB scores between smear positive and negative patients at TB diagnosis (**C**), between the patients requiring standard and extended anti–TB treatment (ATT) at week 3–4 (**D**), month 2–3 (**E**) and month 4–6 (**F**) after treatment initiation are shown in the violin plots. P–values were calculated using the Mann–Whitney U test (ns, $p < 1$; *, $p < 0.05$; **, $p < 0.01$; ***, $p < 0.001$; ****, $p < 0.0001$). (**G**) Scatterplots show TB scores throughout treatment course, color–coded by the patients requiring standard or

extended ATT. The line represents the median of the stratified group with 95% confidence interval around the median shown in the shaded area. (**F**) ROC curves, stratified by different timepoints after treatment initiation, depict predictive performance of the models for discrimination between the patients requiring standard and extended ATT.

total unique 721 genes) were only detected once. This limited commonality among the signatures could be driven by heterogeneity in cohorts or different computational approaches. In comparing our 45-gene set with 30 previously published gene signatures, 25 out of 45 genes were included in at least three published gene signatures, and particularly genes GBP5, C1QB, FCGR1B, DUSP3, and BATF2 have been repeatedly identified in many reports (**S4 Fig**), suggesting the reproducibility of our network-based meta-analysis approach to generate a consistent result. The remining 20 genes, regardless of having only one match or no match to other gene signatures, demonstrate an association to active TB (**Fig 2L–2O**), and may potentially play a role in increasing model sensitivity for long-term TB risk estimation and improving model discrimination between TB disease and viral infection. Along these lines, we demonstrate that the 45-gene set consistently responds to active TB across cohorts versus other conditions (HC, LTBI, OLD, and Tx) (**Fig 2L–2O**). Moreover, we also observe that the subjects with HIV and TB coinfection exhibit a similar expression pattern in the 45-gene set when we stratify the active TB subjects into HIV positive and negative in the heatmap (**Fig 2L–2O**). This demonstrates the potential for applying gene signatures for TB risk estimation in high HIV burden countries, however additional validation studies acquiring datasets with HIV+ individuals will be required to investigate this further.

While building a predictive model to maximize the orthogonal information of the 45 common genes that is associated to the cohort outcomes, we developed three constitutive steps–taking the ratio of the paired gene expressions, feature down-selection, and supervised machine learning in pooled heterogenous cohorts. Feeding the paired gene ratio into the predictive model not only considers first-order interaction between the genes, but also allows accommodation of various data scales generated by different profiling technologies. To produce a generalizable supervised learning model, we maintained cohort-to-cohort variations by not performing data correction on confounding variables. In addition, the model was regressed to two states: ATB and non-ATB containing HC, LTBI, OLD or Tx, which enforces the model maximizing the variation between ATB and non-ATB while considering the difference among the conditions within non-ATB. Finally, ML model benchmarking (**S5 Fig and S3 Table**) and its hyperparameter optimization with a framework of repeated nested cross validation allowed us to identify an ML model generalizable for independent cohorts.

While generating the reduced model, we implemented a feature stability selection approach, a sensitivity analysis to interrogate the likelihood of a feature selected in the model based on randomly generating subsets of a dataset. The list of feature rankings, which refers to the importance of the features contributing to model prediction is displayed in **S6B Fig**. The reduced model is currently considered to use the minimal number of the features that produced sufficient discriminative power as compared to the full model (**S7 Fig**). When applying for a PoC screening test, other factors including selection of the technology, either qRT-PCR [38] or NanoString [53], and the cost of the assay will also need to be considered. Using the feature ranking list as a guidance, we can generate a new model by adding or removing genes based on the needs in a PoC setting. However, in order to be of maximum value in patient care settings, it might be desirable to improve the technology to utilize all genes in the signature rather than reducing the signature to accommodate the limitations of current technology.

In comparing model performance in TB prognosis and diagnosis (**Figs 3 and S8–S11**), we included four previously published models in this report, each of which has reduced the

number of the genes required to the minimum number in order to develop a PoC screening or triage test. Most of the genes selected in the models are strongly associated with active TB [3,5,18,27]. However, when the models were tested in our pooled dataset which was compiled from multiple cohorts, we observed that the TB scores, particularly from non-active TB groups, such as HC and LTBI, showed much larger dispersion in the corresponding distributions than from the active TB group (**S9 and S11 Figs)**. For instance, the scores from Sweeney 3 in the group of LTBI without progression in 2 years exhibited a clear bimodal distribution due to a combination of two different TB score levels generated from the different cohorts (**S9A Fig)**. As a result, we observed weak discrimination between the non-progressing and progressing groups. BATF2, a single gene model, demonstrated accurate discrimination between active and latent TB [20,27], however, showed low discriminatory power in a multi-cohort setting with different confounding variables (**S11E and S11FFig** ) (similar cases in other models can be also found in**. S11 Fig**). This collective evidence via a meta-analysis suggests that the minimized gene signatures may respond not only to active TB disease, but to other diseases or physiological conditions. As Mulenga et al highlighted [37], individuals with respiratory viral infection had elevated RISK 11 gene signature scores which made it difficult to differentiate between active TB and viral infection. Our full model showed less dispersed distributions in non-active TB groups (**Fig 3E**) and provided strong discrimination between ATB and other groups, including viral infection. Our reduced model, using a part of the full model gene signature, maintained accurate discrimination between ATB and non-ATB, but its performance was reduced against viral infection and other lung diseases (**S10 Fig**). Taken together, the results highlight that models only considering the minimized gene set (the best predictors) may not be sufficient to deal with cohort heterogeneity. A model containing a combination of genes with both strong and moderate predictors may improve model reliability in the context of heterogeneity and gain model specificity to TB.

From TB treatment studies, our gene signature model demonstrates that TB score dynamics correlate with treatment duration (**Figs 5A and 6A**), although patient-specific variations are present (**S12 Fig**), which may result from varied pathophysiological conditions at baseline [54]. Additionally, the values of the TB scores in our model are associated with lung inflammation during treatment (**S15 Fig**). This suggests that the TB scores as a biomarker of host immune status respond to *Mtb* bacterial load and its changes ongoing in the lung environment. Treatment adherence is one of the confounding factors to evaluate treatment outcomes (cure and failure). In the Catalysis study, by dissecting different confounding factors, we demonstrated that both drug adherence and *Mtb* bacterial load at baseline, captured by TB score, were individually associated to treatment outcomes (**Fig 5D–5E**). It is possible that the signature that predicted treatment failure for these poorly adherent patients, before treatment was initiated, was able to identify a metabolic state that correlated to high bacterial load and thus a likelihood of poor outcome. This has also been observed in a study specifically aimed at metabolic differences of TB patients [55]. As a caveat, the sample size for the subjects with poor adherence and also for the treatment failure is small and additional studies are required to validate this result.

We demonstrate complete model development beginning with the collection of massive study transcriptomic datasets, identifying a common TB gene signature, generating a generalized predictive model, and systematically validating the model in a real-world setting. This work paves the way for us to further investigate the relationship between TB score and clinical characteristics, outcomes, and other biomarker measurements in our TB treatment clinical trial studies, and to refine our predictive modeling work with other biomarkers to support future clinical trial development and patient care in general.

## Materials and methods

### Transcriptome datasets

We collected 37 publicly available transcriptome datasets from NCBI GEO and EMBL ArrayExpress from previously published studies (**Tables 1 and S1**). Among the datasets, there were 27 studies containing one or multiple clinical TB states including HC, LTBI, ATB, and treatment (Tx) or other lung diseases (OLD) (datapoints $n = 2914$), and 10 progression or treatment studies with longitudinal timepoints (datapoints $n = 1281$). Of these, 26/37 datasets were derived from microarray experiments and 11/37 were derived via RNAseq. We used 27 out of 37 studies (most are the cross-sectional studies) as the discovery datasets to build the networks for our common gene signature discovery, and to train/test the ML model. The remaining 10 longitudinal cohorts as the validation datasets were used to independently validate prediction performance of the ML model on TB prognosis, diagnosis, and responses to TB drug treatment. To further test model specificity to TB disease, we collected additional 20 microarray datasets from GEO (datapoints $n = 2095$) from patients across 12 countries with multiple respiratory viral infections (**S1 Table**).

### Processing of microarray datasets

Microarray data were downloaded from GEO using the function `getGEOData` from the MetaIntegrator package v2.1.3 with option `qNorm=FALSE` [56]. Missing values in each expression matrix (probes x individuals) were replaced with 0 before performing quantile normalization with the package preprocessCore v1.55.2. Rows were reduced for each dataset by retaining only a single probe for each gene (the probe with the highest gene expression sum across all samples in the matrix). This transformed and filtered expression matrix was used for downstream analysis.

### Processing of RNAseq datasets

RNAseq raw data were downloaded from NCBI Sequence Read Archive (SRA) and processed through our customized bulk RNAseq pipeline. Fastp v0.20.1 [57] with option `qualified_quality_phred = 20` was used to perform quality control, adapter/tail trimming and read filtering on FASTQ data, followed by Salmon v1.4.0 [58] with options `recoverOrphans = true, gcBias = true, and seqBias = true` for quantification of gene transcript expression from the sequence reads. The trimmed and filtered reads were mapped to the human genome Ensembl GRCh38 (release 102). A read counts matrix was generated from the pipeline for downstream analysis.

### Differential gene expression analysis

We performed pairwise differential gene expression analysis on the 27 discovery datasets used in the network analysis. To run differential gene expression between two groups within a given dataset, we required that at least three individuals belong to each group. Four pairwise comparisons were considered for each dataset: (1) ATB vs. HC, (2) ATB vs. LTBI, (3) ATB vs. OLD, and (4) ATB vs. Tx.

   **Microarray data.** We used the limma package v3.46.0 to create the contrasts matrix (using functions `lmFit` and `makeConstrasts`) and fit a linear model for each probe (gene) [59,60]. Finally, we used the functions `eBayes` with option `proportion = 0.01` and `topTable` with option `adjust = "fdr"` to compute the log-odds of differential expression for each gene and correct the p-values for multiple hypothesis testing by the Benjamini-Hochberg (BH) method.

**RNAseq data.** We used `calcNormFactors` from the edgeR package v3.32.1 to calculate the normalization factors, then filtered out lowly-expressed genes from each matrix (retaining genes which had at least 1 count-per-million in at least N/1.5 individuals, where N is the number of individuals) [61]. We then used `voom` from the limma package to transform our dataset before creating the contrast matrix. The remaining steps for linear modeling and expression variability estimation are described previously in microarray data.

## Building gene covariation networks

We constructed four gene covariation networks corresponding to four pairwise comparisons as follows: (1) ATB vs. HC, (2) ATB vs. LTBI, (3) ATB vs. OLD, and (4) ATB vs. Tx.

For each comparison, we collected $M$ datasets for each of which we ran a differential gene expression analysis between the groups. From the results, we retained only genes that had both, (1) a corresponding false discovery rate of 5% or less (BH adjusted p-value < 0.05) and (2) an effect size > 1.5-fold ($logFC > log_2(1.5)$). For each dataset, we then constructed a $1 \times N$ vector that presented $N$ genes and contained the $logFC$ for each gene that passed the above filters or 0 otherwise. We combined vectors from $M$ datasets to generate an $N \times M$ matrix and convert the matrix to +1 if the corresponding $logFC > 0$, 0 if the corresponding $logFC = 0$, or −1 if the corresponding $logFC < 0$ to create matrix $X$ which held information corresponding to which genes were significantly upregulated and downregulated across all datasets (**Fig 1A**).

We constructed a covariation network using NetworkX v2.6.3 (https://networkx.org/) with $N$ nodes (each node represents a gene). Edges between nodes were weighted by considering the covariation between each pair of genes in $X$. We calculated the dot product between each pair of genes (a pair of rows in $X$) to generate a distribution ($D$) of edge weights (range—$M$ to $M$, $M$ = number of datasets). To filter out low-value non-zero edges that may have occurred by chance, we performed a permutation test on $X$. We constructed a null distribution of edge weights by randomly shuffling the columns of $X$ within each row (**S1 Fig**). We repeated this process twenty-five times to create the null distribution ($D'$). For each distribution, $D$ and $D'$, we computed the proportion of edge weights to the left of each edge weight for $t \in \{-M, 0\}$ (normalized by all negative edge weights) and the proportion of edge weights to the right of each edge weight for $t \in \{0, M\}$ (normalized by all positive edge weights). Let $t$ be the proportion of edge weights computed from $D$ and $t'$ the proportion of edge weights computed from $D'$. A threshold $T_l$ for edge weights < 0 and $T_r$ for edge weights > 0 was determined by the exact p-value $\frac{t}{t'} < 0.05$. The edges within $T_l$ and $T_r$ were then excluded in the network.

## Identifying common genes among the networks

First, the weighted degree of a node, the sum of the edge weights of a node, was calculated to represent the degree centrality of a gene in the network (**S2 Fig**). A gene with higher degree centrality indicates its response to ATB more consistently across the cohorts. To look for the most common genes in the network, we modeled the weighted degree distribution based on the assumption that the covariation network is a scale-free network [62] where the degree distribution follows a power law (**Fig 2A–2D**). Therefore, a power-law distribution (a generalized beta distribution) is used to estimate the weighted degree distribution $W$ and the cumulative distribution function (CDF) of $W$ was also estimated. For each network, we identified the genes within the top 5% of $W$. Then we collected the top-ranked genes that occurred in at least two of the networks (**Fig 2E**). Next, we excluded genes with $abs(mean(logFC)) < 0.5$. The final common gene list was composed of 45 genes, where 41 genes were up-regulated and 4 genes were down-regulated in ATB relative to other disease conditions.

## Pathway enrichment analysis

To conduct pathway enrichment, we used REACTOME pathway modules taken from MSigDB (https://www.gsea-msigdb.org/gsea/msigdb/). We ran our 45-candidate gene set through Enrichr [63] as implemented in Python through the package GSEApy v0.10.1 (https://github.com/zqfang/GSEApy).

## Establishing optimized predictive models

**Data preparation and harmonization.** Before compiling all the discovery datasets, the data were transformed to logarithmic scale and standardized by z-score within each dataset. The compiled matrix was filtered to contain the common 45 gene expression profiles from each cohort. The outcomes (TB disease states) of the cohorts were united into two distinct states from model building: 1 (ATB) and 0 (HC, LTBI, Tx or OLD)

**Feature generation and selection.** To build a predictive model able to handle data generated across different platforms (RNAseq, microarray, or qRT-PCR) without further data transformation, an expression ratio between a pair of genes was used as a feature importing into the model. By considering all pairs of 45 common genes, a 990-dimensional feature vector was constructed. The dimensionality of the feature space was reduced by selecting the discriminant features to obtain a more generalizable and more accurate model. Therefore, two-step feature selection was placed in our framework. 1) Univariate feature selection: mutual information is calculated to measure the dependency between individual feature and the outcome. The top 90% of total features were maintained in the feature space. 2) Multivariate feature selection. We implemented two approaches in our modeling framework for different purposes. LassoLars is a LASSO model, a $l1$-penalized regression model, which generates sparse estimators and whose parameters are estimated by the LARS algorithm [64]. This approach was considered when a predictive model with a full set of the selected features was generated to achieve the best prediction performance. The second approach was LASSO with stability selection [65], where subsampling from the data in conjunction with $l1$-penalized estimation was implemented to estimate the probability for each feature to be selected. For each feature ($k$), the stability path is generated by the selection probabilities $\hat{\Psi}_k^\lambda$ as a function of the regularization parameter ($\lambda$) when randomly resampling from the data, and the probability ($P^*$) of the feature was determined by $P^* = argmax(\hat{\Psi}_k^\lambda)$ (**S6 Fig**). The features were then ranked by the probabilities and down-selected using a pre-defined cutoff (**S7 Fig**). This approach was considered when only a small number of features was allowed for qRT-PCR to adapt for ultimate translation to a PoC platform.

**Machine-learning model optimization.** We aimed to construct a supervised learning model that took the common gene signature and produced a disease score ranged between 0 and 1 to recapitulate the spectrum of TB infection to disease as well as responses to treatment. To do so, we tested seven state-of-the-art ML regression algorithms in scikit-learn v1.0.1 ML library, including random forest, elastic-net, support vector machine, adaptive boosting, partial least squares, multi-layer perceptron, and extreme gradient boosting.

To establish an optimized model without overfitting, we designed a framework of 5-repeated 5-fold nested cross-validations (CV) [66]. In each repetition, the dataset was randomly divided into 5 folds, where we used 80% of the dataset as the training set for model building (inner CV) and used the remaining hold-out dataset as the test set to estimate the true prediction error of the model (outer CV), where the goodness-of-fitness of the model was measured by mean squared error (MSE) and R-squared between the predicted outcome and the actual. Notably, due to the unbalanced outcomes (TB states 0 or 1) in the dataset, each fold was stratified to maintain the similar proportion of each state as the complete dataset. In inner

CV, the model was trained the following way. First, the bottom 10% features were filtered out by using univariate feature selection (as described above) with the whole training set. Secondly, the training set was divided again based on the 5-fold CV strategy. The feature set was further refined using either one of multivariate feature selection approaches (as described above) based on minimization of CV error estimates. Third, to pinpoint the optimal hyperparameters of the learning algorithm, a randomized search in high-dimensional parameter spaces [67] was applied to increase search efficiency as compared to the brute-force search approach, and the final parameter setting was determined by the one generating the minimal CV errors. Finally, the true prediction error of the model generated from the inner CV was estimated using the test set in the outer CV. To benchmark the regression algorithms, both the area under the receiver operating characteristic curve (AUROC) and MSE were used to assess the true error of the model (**S5 Fig** and **S3 Table**).

After the model and its optimal parameters was determined in 5-repeated 5-fold nested CV, the final model was re-trained using the whole discovery dataset (27 cohorts) and its actual prediction performance in progression and responses to treatment were systematically validated by the longitudinal validation dataset (10 cohorts) and viral infection datasets (20 cohorts). In the end, we finalized two models based on different multivariate feature selection approaches–the full model with full-set selected features that maximize prediction performance and the reduced model with reduced features that retains comparable performance (**S4 Table**).

## Model validation in TB prognosis and responses to treatment

For the progression datasets (7/10 validation cohorts), the datasets were directly compiled without further batch correction to mimic the real-world situation where the prognostic triage tool is performed at an individual subject level. AUC ROC was assessed to evaluate predictive performance that discriminated between the progression group in different prespecified intervals to disease ($< 3$ months, $< 6$ months, $< 12$ months, $< 18$ months, $< 24$ months, and $< 30$ months) and the group including LTBI without progression during the 2-year follow-up. Additionally, sensitivity, specificity, and positive/negative predictive values (PPVs/NPVs) were assessed for each of these time intervals, when assuming 2% pre-test probability, based on two parameters—1) the predetermined cutoffs defined by 2 standard deviations (SDs) above the mean of the control group [27] to prioritize PPVs and specificity (**S6 Table**), and 2) the maximal Youden Index from AUC ROC which provides the best tradeoff between sensitivity and specificity (**S5 Table**). Furthermore, to determine the uncertainty of the estimates of model performance metrics, 95% confidence intervals (CIs) were calculated as described in Ying et al [68]. To investigate the changes of the model-predicted TB scores along with disease progression among the cohorts, the data points were binned using mutually exclusive time intervals to disease of 0–3 months, 3–6 months, 6–12 months, 12–18 months, 18–24 months, 24–30 months, and >30 months. For the treatment cohorts (3/10 validation cohorts), AUC ROC was assessed to evaluate the performance of treatment monitoring that distinguished active TB at baseline and at timepoints after treatment initiation. Furthermore, we also examined the prediction of treatment outcomes (cure, failure, and recurrence) using the data at early timepoints (baseline, week 1, week 2, or month 1), assessed by AUC ROC. To study TB score dynamics over the treatment course across cohorts with different sample collection schedules, the data points were bucketed into several representative time points.

## Developing a probabilistic TB risk model

To accommodate the heterogeneity of immune responses to TB progression in different demographic groups, we developed a probabilistic model to incorporate the uncertainty of the

responses associated with the clinical outcomes. We first compiled all the transcriptomic data (37 cohorts) and extracted the data that was on the trajectory of TB progression as well as from HC, LTBI, and ATB. TB scores for each sample were generated using our predictive model. Next, we focused on six groups–HC, LTBI, < 24 month to disease, < 12 month to disease, < 3 month to disease, and ATB. For each group, we modeled TB scores using a best-fitted probability distribution in the following steps. Initially, the data was to fit to various distributions in which SciPy v1.7.3, a Python library, has a list of predefined continuous distributions that can be tested against the data, and the best-fitted distribution with appropriate parameters estimated was identified by using sum of square error between the actual and fitted distributions. Next, given the best-fitted distribution, its probability density function (PDF) and cumulative density function (CDF) were estimated. Therefore, consider TB score as a continuous random variable $X$ with PDF $f_X(x)$, the CDF can be obtained from:

$$F_X(x) = \int_0^x f_X(u)du$$

Since TB score is ranged between 0 and 1, the area under PDF between 0 and 1 must be one.

$$\int_0^1 f_X(u)du = 1$$

For the groups of different time intervals to disease and ATB, we calculated the cumulative probability that TB score takes a value less than or equal to $t$:

$$P(X \leq t) = F_X(t) = \int_0^t f_X(u)du$$

Conversely, for the groups of HC and LTBI, we calculated the cumulative probability that TB score takes a value large than or equal to $t$:

$$P(X \geq t) = F\prime_X(t) = \int_t^1 f_X(u)du$$

We used the cumulative probabilities to estimate the probability associated to each of six groups, given a TB score from individuals. To determine 95% CIs of CDF, the Dvoretzky-Kiefer-Wolfowitz inequality was used to generate a CDF-based confidence band [69].

### Statistics

A two-sided Wilcoxon Rank Sum Test was used for pairwise analyses comparing TB disease scores across any clinically defined groups. The BH method was used to calculate false discovery rate (FDR)- adjusted $p$-values for multiple-comparison post-hoc correction. An adjusted $p$-value of less than 0.05 was considered significant. Spearman's rank analysis was used to test for correlations between variables.

### Code

For RNAseq processing pipeline, we used Nextflow (https://www.nextflow.io/) for analytics workflow development and workflow submission into SLURM, a workload manger, to distribute computing jobs on AWS HPC parallel cluster. Microarray/RNAseq data process and differential gene expression analysis were conducted in R with customized scripts. The remaining works, including network analysis, supervised learning model building and validation, and

probabilistic modeling were then established in Python. The code for machine learning was adapted to the multi-processing framework to allow parallel computing by leveraging AWS EC2 instance with 64 CPUs.

## Supporting information

**S1 Table. Cohort lists used for model training and validation.** This is the complete version of **Table 1**. The first sheet contains TB-related datasets and used for training and validation analyses, along with accession IDs, platforms, and which disease comparisons each dataset was used in. The second sheet contains respiratory viral infection datasets for model validation. (XLSX)

**S2 Table. Pathway enrichment analysis of the common gene signatures.** Results from gene enrichment analysis on 45 candidate genes using REACTOME gene sets. (PDF)

**S3 Table. Comparison of model performance metrics using discovery datasets.** 5-fold nested cross-validation performance metrics among seven selected machine-learning approaches using the pooled discovery datasets (27 cohorts, datapoints $n$ = 2914). Outer CV AUROC presented graphically in **S6 Fig**. (PDF)

**S4 Table. Full and Reduced model parameters.** The parameters of the final models trained by the pooled discovery dataset (27 cohorts). (PDF)

**S5 Table. Prognostic performance of models stratified by time interval to disease using cut-offs from maximal Youden Index.** Prognostic performance of the new models developed in this report and published previously for incipient TB, stratified by time interval to disease, using cut-offs identified by the maximal Youden Index based on the best tradeoff between sensitivity and specificity from each ROC. Positive and negative predictive values (PPVs/NPVs) were calculated when assuming 2% pre-test probability. The performance metrics are presented with 95% confidence interval. (PDF)

**S6 Table. Prognostic performance of models stratified by time interval to disease using cut-offs from 2 SDs above mean of control group.** Prognostic performance of the models developed in this report and published previously for incipient TB, stratified by time interval to disease, using the predetermined cutoffs defined by 2 standard deviations (SDs) above the mean of the control group (latent TB infection without progression during follow-up). Positive and negative predictive values (PPVs/NPVs) were calculated when assuming 2% pre-test probability. The performance metrics presented with 95% confidence interval. (PDF)

**S1 Fig. Network building parameter selection. (A-D)** Null distribution of edge weights from permuting log2(Fold Change) values across cohorts and calculating dot products compared against the actual distribution of edge weights from the matrix of log2(Fold Change) values for a given disease comparison (Methods). Edge weights between -2 and 2 are likely to arise by chance and are discarded during network construction. (PDF)

**S2 Fig. Network degree distribution.** Distribution of degree, weighted degree, and eigenvector centrality for the four different networks constructed from (A) ATB vs. HC, (B) ATB vs.

LTBI, (C) ATB vs. OLD, and (D) ATB vs. Tx disease comparisons.
(PDF)

**S3 Fig. Correlation of 45 gene set between disease condition comparisons.** Pearson r correlation coefficient calculated between mean log2(Fold Change) between each pair of comparisons (ATB v HC, ATB v LTBI, ATB v OLD, ATB v Tx) for the 45 gene set. Correlations were computed by first averaging the values column-wise (across datasets) for the data in each heatmap Fig 2L–2O to calculate at vector of length 45 for each comparison, then calculating the Pearson r correlation (as implemented in scipy python package) between each pair of vectors. Correlation values and corresponding p-values are reported in the correlation matrix below.
(PDF)

**S4 Fig. Heatmap of gene signatures.** Genes included in our 45-candidate gene set (the first column in both panels) as compared to genes in gene sets from 30 previously published gene signatures (2, 43). We exclude the genes (563 out of total 721 genes) only detected in one gene signature and display 155 genes present in at least two gene signatures and 3 additional genes present only in our 45-candidate gene set. The left panel displays genes detected in our candidate gene set while the right panel displays genes detected in at least two gene signatures. The numbers in parentheses indicate the number of studies identifying the gene.
(PDF)

**S5 Fig. Comparison of model performance using discovery datasets.** Head-to-head comparison of model performance generated from 5-fold nested cross-validation among 7 selected supervised learning models using the pooled discovery datasets (27 cohorts, datapoints n = 2914).
(PDF)

**S6 Fig. Feature down-selection by a stability analysis. (A)** The stability path for each feature shown as blue line indicates the probability of a feature being selected from randomly resampling the data as a function of the regularization parameter (λ). Features were ranked by the maximum probability and top 20 features are listed in **(B)**. The top 12 features were selected in our reduced model to minimize the number of features while maintaining CV performance (**S7 Fig**).
(PDF)

**S7 Fig. Feature down-selection of the model based on AUROC with 95% confidence intervals from 5-fold CV.** The features were ranked by the stability of features selected from randomly resampling the data (**S6 Fig**). A forward stepwise selection was used to evaluate the model CV performance by adding one feature at a time starting from the top feature. The final feature set was determined when the breakpoint of the AUROC is reached.
(PDF)

**S8 Fig. Prognostic performance of the reduced model for incipient TB using the pooled longitudinal validation datasets.** The distributions of TB scores generated by the reduced model, stratified by categorical interval to disease, are shown in a violin plot (datapoints n = 1281) (left panel). ROC curves depict prognostic performance for incipient TB, stratified by time intervals to disease ($< 3, < 6, <12, <18, < 24, < 30$ months) (middle panel) and mutually exclusive time intervals to disease (0–3, 3–6, 6–12, 12–18, 18–24, 24–30 months) (right panel). AUC and 95% confidence intervals for each interval to disease are shown.
(PDF)

**S9 Fig. Prognostic performance of the 4 published models.** Prognostic performance of the 4 published models (*3*, *5*, *18*, *27*). **(A-D)** for incipient TB using the pooled longitudinal validation dataset (6 TB progression studies). The distributions of TB scores, stratified by categorical interval to disease, are shown in a violin plot (datapoints n = 1281) (left panels). ROC curves depict prognostic performance for incipient TB, stratified by time intervals to disease ($< 3$, $< 6$, $<12$, $<18$, $< 24$, $< 30$ months) (middle panel) and mutually exclusive time intervals to disease (0–3, 3–6, 6–12, 12–18, 18–24, 24–30 months) (right panel). AUC and 95% confidence intervals for each interval to disease are shown.
(PDF)

**S10 Fig. Diagnostic performance of reduced model for active TB using a pooled dataset of all 57 collected cohort studies.** Cohort study metadata can be found in **Tables 1** and **S1** (datapoints n = 6290). **(A)** The distributions of TB scores, stratified by different TB disease stages (HC, LTBI, and ATB), other lung disease (OLD) and viral infection (VI), are visualized in the violin plots. **(B)** ROC curves depict diagnostic performance of the models. AUCs and 95% confidence intervals for each comparison are also shown. Same analysis for full model shown in **Fig 3E and 3F**.
(PDF)

**S11 Fig. Diagnostic performance of 4 published models.** Diagnostic performance of 4 published models (*3*, *5*, *18*, *27*) for active TB using a pooled dataset of all 57 collected cohort studies **Tables 1** and **S1** (datapoints n = 6290). **(A, C, E, G)** The distributions of TB scores, stratified by different TB disease stages (HC, LTBI, and ATB), other lung disease (OLD) and viral infection (VI), are visualized in the violin plots. (B, D, F, H) ROC curves depict diagnostic performance of the models. AUCs and 95% confidence intervals for each comparison are also shown.
(PDF)

**S12 Fig. Temporal dynamics of TB scores.** The temporal dynamics of TB scores generated by the reduced model, stratified by the time of sputum culture conversion to negative (negativity at day 28, 56, 84 and 168, and no conversion at day 168 [failed]). The dashed line represents the TB scores of each patient responding to treatment over time, and the red line represents the median of the stratified group with 95% confidence interval shown in the shaded area.
(PDF)

**S13 Fig. Predictive performance of the reduced model in recurrent TB disease.** ROC curves depicting predictive performance of the model for discrimination between cured patients with or without TB recurrence within 2 years after treatment completion are shown in different colors, stratified by different timepoints after treatment initiation. AUC and 95% confidence intervals for each interval to disease are also shown.
(PDF)

**S14 Fig. Predictive performance of the 4 published models.** ROC curves, stratified by different timepoints after treatment initiation, depict predictive performance of the models for discrimination between patients with bacteriological cure and those with treatment failure at EOT **(A)** and cured patient with or without TB recurrence within 2 years after treatment completion **(B)**. AUC and 95% confidence intervals for each interval to disease are also shown.
(PDF)

**S15 Fig. The relationship between TB scores generated by the reduced model and in vivo pulmonary inflammation.** The Spearman correlations between TB scores and MGIT culture time to positivity **(A)**, Xpert Ct values **(B)**, total glycolytic ratio activity (TGRA) at three time

points after treatment initiation (Day 0, Day 28 and Day168) **(C-E)** are displayed. The scores at baseline stratified by radiologically persistent or cleared lung inflammation at EOT are shown in the violin plot **(F)**.
(PDF)

## Acknowledgments

The authors appreciate Dr. Jill Winter for concept discussion and general feedback in the manuscript, Justin Zhang for viral infection dataset collection, Dr. Aparna Anderson for feedback in statistical modeling, and Dr. Gerard Tromp for feedback in bioinformatics approaches.

## Author Contributions

**Conceptualization:** Roger Vargas, Nicole Frahm, Wen-Han Yu.

**Data curation:** Roger Vargas, Liam Abbott, Daniel Bower, Mike Shaffer.

**Formal analysis:** Roger Vargas, Daniel Bower, Mike Shaffer, Wen-Han Yu.

**Investigation:** Roger Vargas, Nicole Frahm, Wen-Han Yu.

**Methodology:** Roger Vargas, Liam Abbott, Wen-Han Yu.

**Project administration:** Wen-Han Yu.

**Resources:** Nicole Frahm, Wen-Han Yu.

**Software:** Liam Abbott.

**Supervision:** Wen-Han Yu.

**Visualization:** Roger Vargas, Daniel Bower, Mike Shaffer.

**Writing – original draft:** Roger Vargas, Wen-Han Yu.

**Writing – review & editing:** Roger Vargas, Nicole Frahm, Wen-Han Yu.

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
