## [Decision Letter · Decision Letter 0]

21 Feb 2023

Dear Dr. Yu,

Thank you very much for submitting your manuscript "Common gene signature model discovery and systematic validation for TB prognosis and response to treatment" for consideration at PLOS Computational Biology.

As with all papers reviewed by the journal, your manuscript was reviewed by members of the editorial board and by several independent reviewers. In light of the reviews (below this email), we would like to invite the resubmission of a significantly-revised version that takes into account the reviewers' comments.

We cannot make any decision about publication until we have seen the revised manuscript and your response to the reviewers' comments. Your revised manuscript is also likely to be sent to reviewers for further evaluation.

Sincerely,

Marc R Birtwistle, PhD

Academic Editor

PLOS Computational Biology

Amber Smith

Section Editor

PLOS Computational Biology

Reviewer's Responses to Questions

**Comments to the Authors:**

Reviewer #1: Vargas and colleagues present a network-based meta-analysis approach using whole blood transcriptome data from multiple cohorts to identify a shared gene signature of active TB (ATB) infection. This work's major contribution is identifying a set of genes that can be assembled as a machine-learned signature that discriminates active ATB infection from healthy controls, latent TB, and other diseases across highly heterogeneous cohorts. Until now, no signature has been accurate across studies – this work, therefore, represents a major step forward for the field. They then train two ML models on the pairwise expression ratios of a subset of the identified genes to predict a TB score representing the spectrum of TB disease and treatment response.

MAJOR SUGGESTIONS:

1. It is unclear why a complex network approach to selecting the gene sets is preferred over simply choosing commonly differentially expressed genes and pathways among the heterogeneous studies. Given that the selected genes did not distinguish ATB from OD (or Tx), I wonder whether a set of common over-represented genes may perform better in the ML models.

2. Neither the Catalysis nor the Leicester treatment response cohorts used for validation had HIV-positive patients. The authors should either validate their models’ predictions for this patient population or avoid stating that it could be reliably used for treatment outcomes in countries with high HIV burdens.

3. It is not clear how genes were determined in the reduced model. The motivation for creating the reduced model is briefly discussed near the end of the paper, but the derivation is mostly ignored. Based on the network in Supplementary Figure 4, many of the genes that are chosen in this reduced model are not very heavily connected, which was made out to seem like a critical criterion in the 45-gene model.

4. The writing is difficult to follow – as a result, there are details and nuances I cannot understand or review. In some areas, insufficient information is given (how were the networks assembled? How were the genes determined in the reduced model?). And in other areas, it was difficult to understand what was done and why (the paragraph about PPV and NPV). Paragraphs are long and dense and would benefit from section headings based on conclusions, not processes.

5. Is the reduced gene set predicted to perform better as the number of cohorts is increased?

MINOR SUGGESTIONS:

1. Page 8, paragraphs 1 and 2: In paragraph 1, the authors state they utilize cohorts containing different clinical/genetic/technical covariates to “identify a set of common genes that robustly distinguish ATB from LTBI, HC, OD and ATB treatment.” The distinction between ATB and ATB treatment is made confusing by the use of “undergoing TB treatment (Tx)” to refer to the same group in paragraph 2 and in the rest of the paper. This should be made consistent throughout the manuscript.

2. Page 15, paragraph 2: The authors state – “Taken together, the results highlight that model only considering the minimized gene set […] may not be sufficient to deal with cohort heterogeneity.” In this sentence, “model” should be changed to “models” or “a model”.

3. Figure 2, panel D: the ATB treatment group is abbreviated as “Tret”. Elsewhere in the paper, the same group is referred to as “ATB treatment” and “Tx”. This should be made consistent across the manuscript.

4. The text states that “the full model was able to statistically separate the subjects’ scores between clinically defined subgroups and healthy controls as well as in-between clinically defined subgroups”, however, Figure 3B shows no significance in difference between the subclinical TB and clinical TB groups.

5. The figure text is very small, making it difficult to read. I was unable to review the figures in printed format.

6. The figures should use color more deliberately. Within the same figure, red means no disease (Fig 6A) or severe disease (Fig 6B). It is not clear that color is needed at all in the violin plots, because they are labeled along the x-axis.

7. Consider using fewer abbreviations. Some are nonstandard (OD reads as optical density to experimentalists!) and others (PPV and NPV) are not defined in the main text or in the first instance.

8. Some language should be made more formal, for example “didn’t” appears several times.

9. Please consider adding line numbers – it makes it easier to comment on specific areas.

10. Consider consolidating some of the supp into intuitive figures or tables for the main text. As written, a reader who does not download the supp materials (which is most readers) will have to believe and not see how the models were built and features selected.

11. A table showing how data from the different studies were used in training and testing (or both) would help the reader follow how to evaluate independent tests of the models.

Reviewer #2: Vargas et al., compiled 57 transcriptomic datasets from whole blood of tuberculosis (TB) patients, healthy individuals, or other diseases and used a combination of network modeling and machine learning to generate a new gene signature with the improved ability to distinguish active TB infection from healthy and other viral respiratory diseases. The model possesses an AUROC of 0.85 (74.2% sensitivity and 78.3% specificity) which is essentially in accordance with the WHO target product profile for prediction of progression to TB. Overall, I find the study to be well performed and interesting, possessing clinical relevance for an important global problem. I believe this manuscript warrants publication in Plos Computational Biology. I have detailed a few minor comments below:

Specific comments:

- I would encourage the authors to create a figure panel to support this assertion “The closer to the center (the higher weighted degrees) the nodes are, the more consistently the genes respond to active TB among various studies.” Perhaps by correlating gene log2FCs and gene network centrality across datasets/conditions?

- Some genes, like HP, appears to have extremely variable log2FC across datasets as evidenced from the heatmaps in Figure 2. To me, this warrants its exclusion from the final gene signature. Should the authors also include a requirement for low inter-dataset variability using variance or standard deviation of expression across datasets? I imagine genes with high variability across datasets might not be as reliable for clinical use. I leave it up to the authors to decide whether this would improve their results but encourage them to explore this.

- I would encourage the authors to provide figure evidence for this statement: “In addition, we plotted differential expression of the 45 genes in heatmaps across all datasets (Fig. 2E-H) and observed that gene patterns of upregulation and downregulation were consistent across cohorts within each disease condition and between disease conditions.” For instance, a condition correlation matrix where gene expression values are averaged for each condition and then correlated across conditions, or similar. In summary, it would improve the rigor if the authors provided an additional analysis to prove expression to be similar across conditions, which goes beyond merely displaying the heatmaps.

- I would encourage the authors to move or re-create a gene set enrichment analysis of the 45 genes in the main, perhaps in Figure 2. The authors could also doing this for the smaller ML refined gene signature. The authors could even consider a clustergram with terms along the rows and genes along the column, indicating how each gene maps to each of the top X significantly enriched terms, though I leave the presentation choice to the authors. Presenting this visually the main figures will help the readers better appreciate the biology revealed by the gene signature.

- For the start of the machine learning section, a visual flow chart would be helpful to understand how features were defined and refined, and an indication of the size of the resulting final gene signature. In addition, the final ML refined gene set could be depicted as a heatmap (as in Figure 2) again in Figure 3.

- The respiratory virus control is an essential piece of this analysis. Without it, patients could be misdiagnosed, especially since the signature is composed of many inflammation-related genes. If possible, authors should include respiratory virus control in Figure 3. In general, the authors should include this control for all analyses, when possible. This was a major concern of mine when I approached this work.

- The approach to use network modeling to refine the gene set is an interesting one with general applicability to other datasets and biological areas. However, one could imagine going directly from all genes into the machine learning algorithm or one could imagine alternative methods (e.g. lasso regression) to refine the gene set. At a minimum, the authors should comment on why they felt network modeling would out-perform other approaches (in the Discussion). Ideally, the authors should attempt to prove that incorporating the network modeling, versus excluding it, either (1) improves the predictive power of their resulting gene signature, (2) enhances the feasibility of the analysis, or (3) both.

- Some aspect of the analysis comparing to previously published signatures (Sweeney 3 and RISK 6) could be featured in the main as this will probably be of high interest to scientists in the field.

- In general, figure panels could be smaller and more compact to eliminate white space and maximize the transfer of information from main figures. Many interesting aspects from the supplement could be included in the main, for example, if space was optimized.

- The title could be improved to communicate the most important finding of the manuscript. For instance, it is unclear to me why the word “common” is included in the title and feel it dampens the enthusiasm for the work. As an example: "Forty-five gene transcriptomic signature identifies TB infection across diverse patient cohorts”. In the end, however, I defer to the authors as to their choice of title.

M. Bouhaddou

**Have the authors made all data and (if applicable) computational code underlying the findings in their manuscript fully available?**

Reviewer #1: Yes

Reviewer #2: Yes

PLOS authors have the option to publish the peer review history of their article (what does this mean?). If published, this will include your full peer review and any attached files.

Reviewer #1: No

Reviewer #2: No
---

## [Decision Letter · Decision Letter 1]

15 Jun 2023

Dear Dr. Yu,

We are pleased to inform you that your manuscript 'Gene signature discovery and systematic validation across diverse clinical cohorts for TB prognosis and response to treatment' has been provisionally accepted for publication in PLOS Computational Biology.

Best regards,

Marc R Birtwistle, PhD

Academic Editor

PLOS Computational Biology

Amber Smith

Section Editor

PLOS Computational Biology

Reviewer's Responses to Questions

**Comments to the Authors:**

Reviewer #1: This revision addressed my concerns from the first review. A final suggestion is to try to significantly increase the size of the fonts - in many cases, more letters are used than needed (a lot of redundancy) - if eliminated, the figure could be enlarged to be publication quality. This will encourage more people to read this excellent work.

Reviewer #2: Congrats on a cool study!

Reviewer #3: In this manuscript, the authors use network-based meta-analysis to integrate diverse transcriptome datasets to identify a common 45-gene signature specific to active tuberculosis (TB) disease across studies. The authors then proceed to show that this TB-specific gene signature model has significant predictive power in both TB risk estimation and treatment monitoring. Overall, this is a well-written paper addressing an important problem of TB gene signature development. The developed TB gene signature is shown to be highly useful in a variety of settings, ranging from disease risk estimation to treatment monitoring.

I have carefully examined the authors' response to previous reviewers' comments. I find that the authors have adequately addressed all comments from previous reviewers through revisions. Hence, I recommend accepting the revised manuscript for publication in its present form.

**Have the authors made all data and (if applicable) computational code underlying the findings in their manuscript fully available?**

Reviewer #1: Yes

Reviewer #2: Yes

Reviewer #3: Yes

PLOS authors have the option to publish the peer review history of their article (what does this mean?). If published, this will include your full peer review and any attached files.

Reviewer #1: No

Reviewer #2: No

Reviewer #3: No

---

## [Editor Report · Acceptance letter]

10 Jul 2023

PCOMPBIOL-D-22-01737R1 

Gene signature discovery and systematic validation across diverse clinical cohorts for TB prognosis and response to treatment

Dear Dr Yu,

I am pleased to inform you that your manuscript has been formally accepted for publication in PLOS Computational Biology. Your manuscript is now with our production department and you will be notified of the publication date in due course.

With kind regards,

Zsofi Zombor
